# Global implications of uncertainty in China's climate policy delivery

**Dan Zhang** [1] ✉, **Steve Pye** [1], **Jim Watson** [2], **James Price**[1] & **Dan Welsby** [2]

The delivery of China's climate policy has substantial global implications, yet persistent gaps between policy targets and implementation raise concerns about policy credibility. This study presents a structured credibility assessment of 292 targets across 58 national climate and energy policies, including China's updated 2035 NDC. We apply a morphological scenario framework to examine interactions between policy uncertainty and socio-economic and technological drivers, embedding these scenarios in an integrated assessment model under two global contexts: one aligned with current NDCs and one consistent with global net-zero ambition. We find that timely or accelerated delivery of China's net-zero target could partially offset insufficient ambition elsewhere by mid-century. Full delivery by 2050 reduces global $CO_2$ emissions to 13 Gt, compared with 23 Gt without such policies. Policy delivery uncertainty alone could imply up to ~500 $GtCO_2$ difference in cumulative emissions by 2100 (~0.17 °C warming).

Recent years have seen substantial changes in China's climate governance, reflecting a growing effort to align decarbonisation goals with economic development. Following its updated Nationally Determined Contribution (NDC) targets in 2021, China incorporated carbon reduction into national planning for the first time through the 14th Five-Year Plan (2021–2025)[1,2]. The introduction of the "1 + N" policy framework, combining a national carbon neutrality roadmap ("1") with sectoral and regional action plans ("N"), has since guided coordinated implementation across multiple levels of governance[3,4]. In September 2025, China further updated its NDC for 2035, introducing for the first time an absolute greenhouse gas (GHG) emissions reduction target of 7–10% below the peak level by 2035.

As the 14th Five-Year Plan is in its final year in 2025, persistent gaps between central government directives and on-the-ground implementation (SI Table 3) raise critical questions about the effectiveness of China's climate policy architecture and the credibility of its multi-level climate targets. As the world's largest emitter, accounting for 32% of global $CO_2$ emissions in 2024, and the largest consumer of coal and leading importer of oil and gas, the implications of China's policy delivery extend well beyond its national borders[5]. In the absence of international ambition - especially as major economies like the US stall

or roll back their commitments, understanding the role and limits of unilateral action becomes essential.

Our analysis of China's climate policy delivery proceeds in three steps. First, we develop a systematic assessment of the credibility of China's climate and energy policy targets. While policy credibility has been increasingly discussed in the academic literature, it remains weakly defined and rarely quantified in a systematic way[6–9]. We adapt the framework proposed by Rogelj et al. to suit the characteristics of China's political system[6]. A total of 292 numerical targets across 58 policy documents are reviewed. Of these, 47 quantifiable targets are filtered and implemented in the energy system model (see Methods). Target's credibility is evaluated using three indicators: (i) the governance level at which it is set, (ii) whether it is explicitly stated in national Five-Year Plans, and (iii) the degree to which it is on track to be achieved. These indicators are combined into a weighted credibility score that reflects varying levels of policy credibility, ranging from targets that have already been achieved to those assessed as low credibility.

These differences in delivery reflect deeper structural trade-offs in China's climate policymaking, particularly the tension between long-term net-zero commitments and near-term socio-economic

[1]UCL Energy Institute, University College London, London, United Kingdom. [2]UCL Institute for Sustainable Resources, University College London, London, United Kingdom. ✉e-mail: dan.zhang.22@ucl.ac.uk

development priorities. China's case is distinctive due to the government's central role in shaping social and economic development, including setting specific growth targets and implementing population planning policies. This strong state-led approach means there is an interplay between broader system-level changes (such as GDP and population) and domestic policy delivery, which Shared Socioeconomic Pathways (SSPs) tend to overlook[10]. As a result, most global integrated assessment model (IAM) projections of energy service demand that rely on SSP-based drivers lack adequate country-specific diversity.

To address this gap, the second stage of our analysis employs a morphological scenario framework (see Methods) to construct internally consistent narratives of China's energy future, which are then assessed using the TIAM-UCL. We develop four scenarios that reflect the interplay between policy implementation and broader system-wide factors, including economic development, demographic trends, and technological innovation. The scenarios capture a diverse set of futures for China's climate governance, alongside corresponding shifts in socioeconomic and technological development (see Methods and SI section 2). The Great Wall (GW) reflects a constrained future marked by economic stagnation, declining population, and stalled climate progress. Red Sun (RS) envisions strong economic growth, driven by real estate market recovery and urbanisation, while climate goals are deprioritised. Calm Sea (CS) presents a balanced pathway where China maintains moderate economic growth while fully implementing current climate policies. Green Lights (GL) depicts a future where China demonstrates climate leadership while maintaining high economic growth. This range of scenarios reflects increasingly effective delivery of sectoral and national policy targets, informed by the credibility assessment.

Our modelling indicates that China's current policy framework has set its energy system on a path of accelerating electrification, driven by credible efforts to expand EV adoption and renewables. However, the lack of explicit, enforceable fossil fuel phaseout targets, especially for coal, continues to undermine climate goals. Ambiguities in industrial policy and weak enforcement indicate a persistent misalignment between national targets and local growth incentives, weakening sectoral credibility. Achieving net-zero before 2060 could require electrification rates above 48%, variable renewables supplying nearly 68% of electricity, and substantial electrification and substantial reductions in coal use in the industrial sector. Our results provide important insights to inform the design of China's next Five-Year Plan and highlight the need for the central government to clearly assign responsibilities across regions and strengthen incentives for credible local implementation.

Finally, in our third step, we quantify the global impact of China's policy uncertainty by embedding our four national scenarios into two global contexts, either limited to achieving current NDCs or delivering the net-zero target (see Methods). This framework highlights China's potential role in global decarbonisation, providing a clear assessment of its relative contribution to such a target. Our modelling suggests that differences in China's climate policy delivery alone lead to cumulative global emission variation of up to around 500 GtCO$_2$ by 2100, which translates to increased or decreased warming of up to 0.17 °C. While a more ambitious domestic policy could see China compensating for the failure of other nations to fully deliver on their mitigation objectives by mid-century, it cannot fully offset prolonged global inaction.

This study contributes to the literature in three key aspects. First, compared to Chinese government official assessments of policy progress, our structured credibility assessment offers an evaluation of climate and energy targets, uniquely linked to integrated assessment modelling. In contrast to most China-focused scenario studies, which adopt normative assumptions and lack consistent socioeconomic narratives, our approach systematically decodes national policy

rhetoric into four exploratory scenarios reflecting China's development priorities and constraints[11-13]. Our study evaluates the potential implications of China's new 2035 NDC for its energy transition and global decarbonisation using an energy modelling framework. Second, it bridges a critical gap between global IAM research and detailed China-specific modelling of policy scenarios, providing a structured approach to assess the role and limits of unilateral action under varying global contexts[14,15]. Third, it offers empirical insights into China's policy implementation challenges and offers actionable evidence to inform future national governance and international cooperation.

## Results

### Chinese decarbonisation and energy transformation
**Existing policy efforts ensures high certainty in electrification and renewable expansion.** Our scenario analysis indicates that existing policy efforts over the past five years have driven system-wide electrification and renewable power expansion (Fig. 1a, b). As policies progressively strengthen from the Great Wall (GW) to the Green Lights (GL) scenario, and as more high-level climate objectives are assumed to be successfully achieved, the electrification rate (share of electricity of final energy consumption) rises to 48% under GL by 2050, nearly doubling the 2020 level (Fig. 1a). While current power sector policies lack explicit coal control targets, China's overachievement with respect to wind, solar capacity installation targets, combined with declining costs, has accelerated and could continue to accelerate renewable deployment. Even in the most conservative scenario (GW), coal could phase out by 2060, while wind and solar generation reach 7.6 PWh by 2060−eight times the 2020 level.

However, the pace and depth of power sector decarbonisation are still strongly shaped by uncertainty in national net-zero target achievement. Compared with the Calm Sea (CS) and GL scenarios, achieving the net-zero target requires an immediate reduction in coal generation from current levels and a more ambitious scale-up of wind and solar generation, reaching 14.8 PWh in CS and 17.5 PWh in GL, both accounting for around 68% of total electricity generation by 2060 (Fig. 1b). Our sensitivity analysis suggests that to achieve the 2035 GHG reduction targets, coal generation must be phased out more rapidly, resulting in an additional ~1 PWh annual reduction in 2040 compared with the main cases of RS and CS (see SI Fig. 5a)

**Policy credibility differences create uncertainty in industrial decarbonisation.** Although the expansion of electrification improves system-wide energy efficiency, existing efforts in renewable power and electric vehicle (EV) alone is insufficient to deliver deep decarbonisation due to the weakness of industrial policies. This weakness is reflected in two aspects: the limited credibility of current industrial measures (e.g., steel scrap use, energy intensity reduction) and the absence of explicit coal reduction targets in the current policy landscape.

Without binding net-zero commitments in GW and RS, industrial coal consumption is estimated to grow to 37− 40 EJ per year by 2060 to support varying degrees of economic development (Fig. 1c). Under net-zero ambition in CS and GL, industrial coal is constrained via accelerated electrification but still requires steep post-2050 reductions, placing increased pressure on coal phaseout in the petrochemical sector and driving increased demand for alternative fuels such as bioenergy and hydrogen (Fig. 1a). Taken together, the attainment of China's national net-zero goal is fundamentally dependent on a substantial industrial electrification and coal reduction. Therefore, clearer coal phaseout targets and stronger enforcement of industrial decarbonisation should be central to China's next phase of climate governance.

**Policy ambiguity heightens the risk of oil and gas lock-in in the energy system.** Compared to coal, oil and gas play limited but

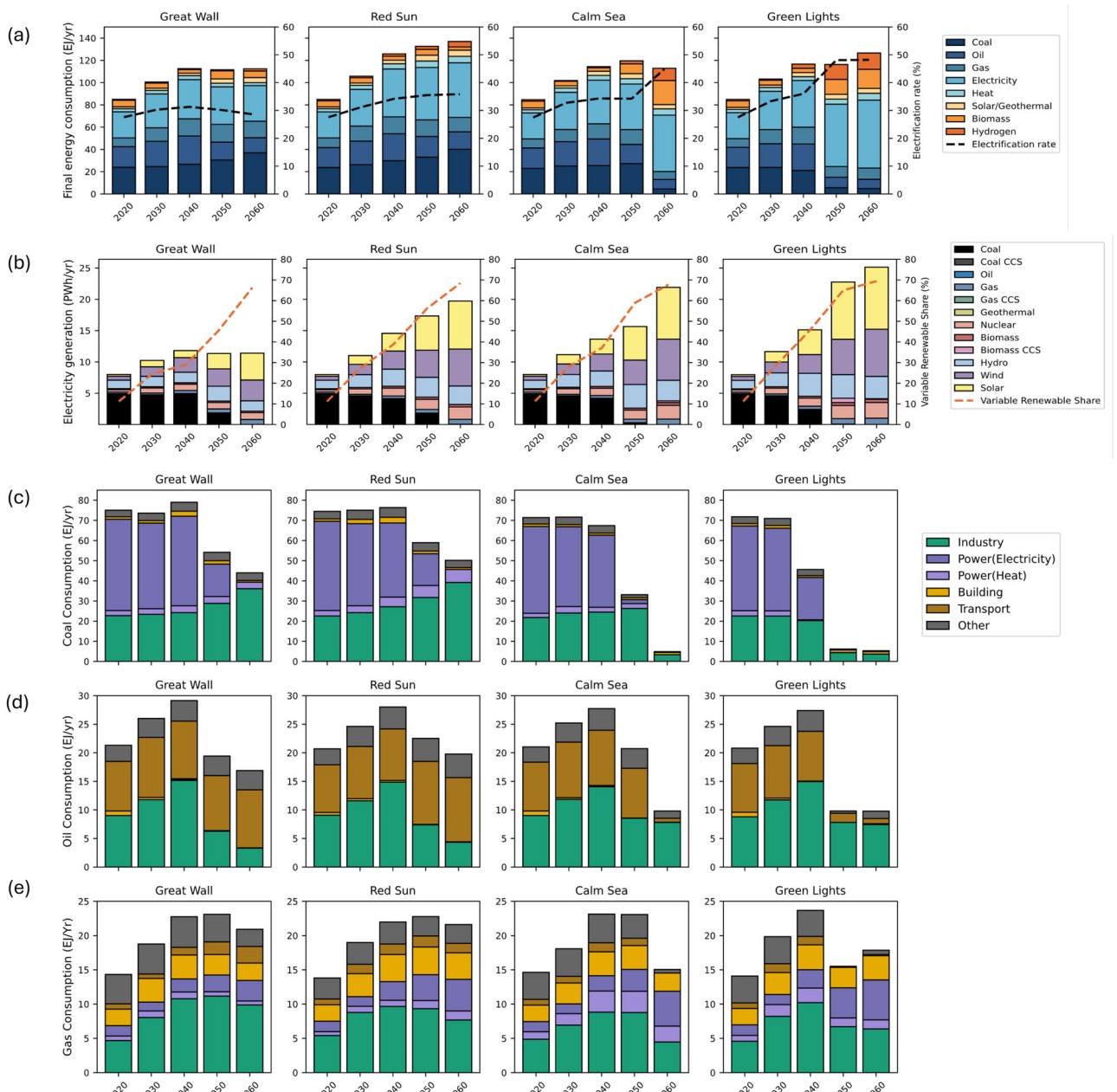

**Fig. 1 | Chinese energy transition metrics across selected scenarios. a** Chinese final energy consumption across scenarios, with dashed trend line representing total electrification rate (measured as the share of electricity in final energy consumption across end-use sectors.) **b** Electricity generation mix across scenarios, with dashed trend line representing variable renewable energy (VRE) share. Panel c-e: Fossil fuel consumption by sector across scenarios; **c:** coal; power sector in TIAM-UCL includes both electricity generation and heat production; coal consumption

observed in GW and RS in 2060, without corresponding coal power generation, represents coal used for heat production. **d** oil, and **e** gas, sector 'Other' refers to upstream and agriculture in TIAM-UCL. It should be noted that the China-related modelling results presented in Fig. 1 are based on the global NDC context. However, because of the underlying TIMES model structure, China's domestic energy transition is primarily driven by end-use demand and domestic policy constraints, and is not materially affected by assumptions about the rest of the world.

persistent roles in China's energy system over the long term, mainly as chemical feedstocks. Overall, oil and gas never exceed 20.8% (RS) or fall below 14.5% (GL) of the total energy mix, and neither is fully phased out, even under net-zero scenarios, highlighting the challenge of decarbonising hard-to-abate sectors.

Oil demand in China is primarily driven by the transport and industrial sectors. Given the country's notable progress in EV adoption, high-credibility EV policies are assumed to be implemented at an increasing rate from GW to GL, leading to lower oil use in road transport. However, total transport-related demand remains uncertain due

to non-road sectors such as shipping and aviation. In the net-zero scenarios (CS and GL), oil is nearly fully phased out from transport by the target year, positioning the sector as the greatest source of oil reduction but also intensifying pressure to scale up bio jet kerosene in aviation and hydrogen and ammonia fuels in shipping (Fig. 1d). However, lingering industrial demand, particularly for petrochemical feedstocks, remains the final barrier to a complete oil phaseout.

Natural gas demand indicates distinct sectoral and temporal patterns across scenarios, shaped by various drivers. While the government has not set explicit sectoral targets, it prioritises residential

and public service use, allows the use for peak power generation, and restricts natural gas industrial use of methanol and urea production[16]. This could result in increased seasonal and hourly fluctuations in gas demand. Given TIAM's limited temporal resolution, our results indicate that annual gas demand in the buildings sector remains relatively stable, ranging from 2.4 to 3.5 EJ (15%–20% of total use). In GW and RS, additional demand comes from gas-fuelled trucks and LNG bunkering, reflecting current market patterns. In contrast, net-zero scenarios (CS and GL) require a complete transport phaseout and gradual reduction in industrial levels from 2040 onward. Total gas use peaks at ~23.6 EJ (620 bcm) around 2040 if net-zero targets are achieved, as observed in CS and GL (Fig. 1e).

Notably, China's heavy reliance on imports (73% for oil and 40.9% for gas) adds complexity to energy policy making and shapes government perspectives on the strategic role of these fuels[17]. Our policy assessment finds that government attention is primarily focused on objectives related to energy security and system stability, such as domestic production targets, pipeline infrastructure, and storage facilities. These priorities are reflected in national planning and most policy targets are achieved by 2025 or are rated as highly credible (see SI Table 3). This focus reflects an ambiguous stance on the long-term phaseout of oil and gas, increasing the risk of their continued lock-in to the energy system. Given that China is the world's largest oil and gas importer, its future demand trajectory carries important implications for the pace of the global fossil fuel phaseout.

### Global Implications of Uncertainty in China's Climate Policy Credibility

Panels a and b in Fig. 2 indicate that China plays a varying but consistent role across different global climate ambition contexts. For a global net-zero context, annual $CO_2$ emissions decline from approximately 40 Gt $yr^{-1}$ in 2020 to between ~23 Gt $yr^{-1}$ and ~13 Gt $yr^{-1}$ by 2050, depending on uncertainties in China's policy delivery. Whether China prioritises economic growth without strong decarbonisation (RS) or pursues green development (GL) shapes its contribution to global climate outcomes. In the RS scenario, China's share of global annual CO2 emissions reaches 32%–42% by 2050, posing a major challenge to global decarbonisation. In contrast, the CS scenario sees China's share fall from 32% in 2024 to 21%–30% by 2050 with timely net-zero delivery. If China were to achieve net zero even earlier, by 2050, as in GL, its share would drop to zero, firmly positioning it as a global climate leader.

Within each global context, uncertainties in China's domestic policy pathways lead to a substantial spread in global cumulative net $CO_2$ emissions and associated warming outcomes (Fig. 2c, d). By 2050, cumulative emissions differ by ~75 $GtCO_2$ in both the NDC (1031–1106 Gt) and Net Zero (1006–1081 Gt) groups. By 2100, these gaps will widen further. Cumulative emissions range from 1909 to 2421 $GtCO_2$ under the NDC context and from 1153 to 1634$GtCO_2$ under Net Zero, representing a potential 500 $GtCO_2$ difference across Chinese scenarios. This magnitude is almost half of the IPCC-estimated 1150 $GtCO_2$ global carbon budget for a > 50% chance of limiting warming to 2 °C from 2020 onwards, underscoring the scale of China's global climate leverage. Achieving the additional 2035 GHG emissions reduction targets set out in China's updated NDC could lower global cumulative $CO_2$ emissions by approximately 4–6 Gt (7% reduction case) and 6–10 Gt (10% reduction case) across our scenarios (See SI Table 14).

Correspondingly, estimated warming from TIAM-UCL diverges more substantially, ranging from 2.46–2.63 °C under the NDC context and 2.05–2.23 °C under Net Zero. These results demonstrate that internal uncertainties within China, particularly regarding the credibility and delivery of its climate policies, as well as associated societal and technological transitions, could shift global warming outcomes by as much as 0.17 °C by the end of this century.

However, whether China's ~0.17 °C of climate influence alone is decisive in determining whether global warming stays below 2 °C still depends on the level of climate ambition in the rest of the world. Our results indicate that by mid-century, China's domestic actions can play a pivotal role in narrowing the ambition gap: emissions trajectories under the NDC and Net Zero contexts partially overlap depending on China's policy delivery (Fig. 2c). This suggests that China has the capacity to buffer or delay the global consequences of insufficient action elsewhere—at least temporarily.

Nevertheless, China alone cannot fully offset the cumulative emissions and temperature differences that result from divergent global efforts by 2100. Even under China's most ambitious scenario (GL), cumulative emissions and warming outcomes under the NDC context remain higher than under the net-zero context, underscoring the structural limits of unilateral action, even from the world's largest emitter today, in the face of global inaction. Therefore, in a world where the remaining carbon budget is rapidly shrinking, the credibility and timely delivery of China's internal climate policies are essential but must be complemented by stronger international ambition if key temperature thresholds are to be preserved.

## Discussion

Our analysis shows that China's climate policy credibility plays a decisive role in shaping both the pace and structure of its energy transition. Preliminary progress of electrification and the power sector transition in China has been driven by sectoral targets in transport, buildings, and renewable capacity expansion, with most assessed as 'achieved' or 'high' in credibility. These existing policy efforts provide a high degree of certainty that electrification expansion and renewable adoption will continue, supported by clear goal-setting, strong governance, and consistent inclusion in successive Five-Year Plans. Notably, many of these are incremental policies that enhance political feasibility by allowing phased, adaptive implementation.

While electrification has been a central focus of China's policies and has driven early progress, it is not sufficient on its own to achieve long-term net zero. When comparing scenarios that include China's most credible polices (GW and RS) with scenarios that also include stated ambitions (CS and GL), our analysis highlights persistent gaps between ambition and implementation. Net-zero alignment (CS and GL) requires an immediate reduction in coal-fired power generation and substantial electrification combined with a reduction in coal consumption in industry (Fig. 1b, c). Achieving the 2035 GHG emissions targets likewise underscores the importance of coal phaseout. The current delay in this particular transition reflects, on the one hand, a persistent policy gap between sectoral transitions and overarching climate pledges, and on the other hand, the low credibility of relevant policies, undermined by weak enforcement.

The policy gap is particularly evident in the absence of explicit, enforceable sectoral targets for fossil fuel phaseout. Vague policy signals such as "strictly limit" coal consumption or "strictly control" new coal projects weaken target credibility and allow discretionary interpretation at the local level[1]. Meanwhile, the prioritisation of domestic oil and gas production for energy security sends mixed signals, potentially reinforcing fossil fuel lock-in and diluting the consistency of China's decarbonisation agenda (See SI Table 3). Further work is needed to examine how a more serious approach to coal phaseout could ensure system reliability, especially by scaling up viable alternatives for maintaining power system balance as coal declines.

Our policy review suggests that the weak enforcement partly stems from ambiguity in central policy design and deeper structural tensions in China's climate policymaking and governance. For instance, industrial efficiency regulations define both basic and advanced performance standards but apply only to enterprises above a designated size, narrowing regulatory scope[18]. Even within the

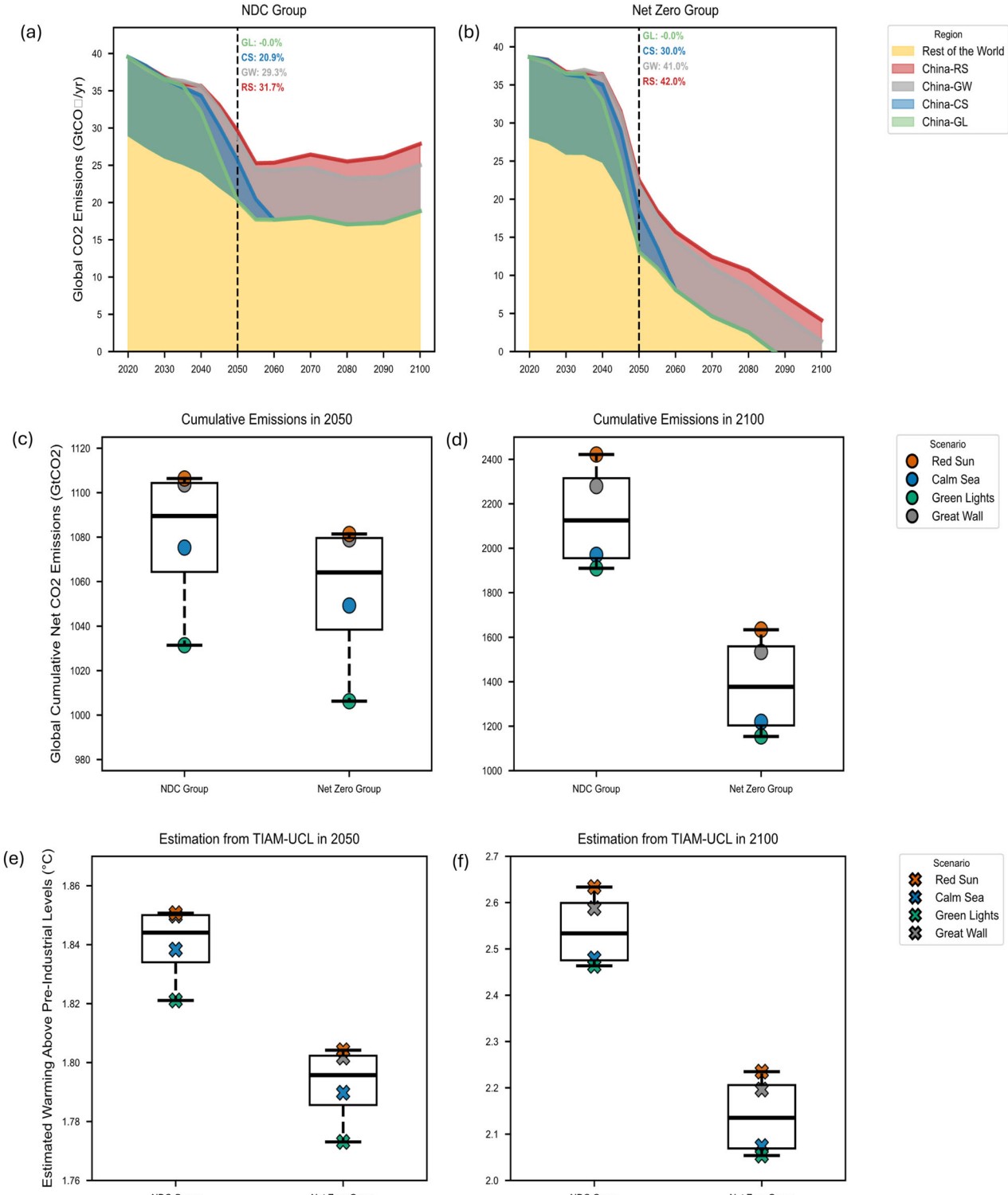

**Fig. 2 | Global carbon emission and average temperature increases under differentiated international climate ambition levels (NDC and net zero) and Chinese climate policy delivery outcomes.** Panels **a**, **b**: Global carbon emissions by scenarios assuming either a) NDC or b) Net Zero ambition for the Rest of the World, with the percentage values reporting China's share of global carbon emissions in 2050; Panels **c**, **d**: Global cumulative net carbon emission in **c**) 2050 and **d**) 2100 across scenarios, assuming either NDC or Net Zero ambition for the Rest of the World; Panels **e**, **f**: Global average temperature increases in **e**) 2050 and **f**) 2100 across scenarios, assuming either NDC or Net-Zero ambition for the Rest of the World. Box plots summarise outcomes across the four China scenario families within each global policy framework (*n* = 4 per group; Great Wall, Red Sun, Calm Sea, and Green Lights). Centre lines indicate medians, boxes show the interquartile range (25th–75th percentiles), and whiskers extend to 1.5× the interquartile range. Individual points represent results for each scenario family (one point per scenario).

regulated group, the required share of firms meeting advanced standards rarely exceeds 50%, reflecting accommodation of regional development disparities (See supplementary data). Many enterprises continue to struggle to meet even the basic thresholds. This lack of clarity enables local governments to selectively interpret and implement policies based on local economic priorities.

However, due to the quantification limits of TIAM-UCL, targets related to water, waste, recycling, air pollution, and climate finance are excluded from our modelling (see SI Table 2). Nonetheless, the 47 modelled targets cover all major energy-related sectors, including electricity and heat production, transport, industry, and buildings. Together, these sectors account for around 96% of China's $CO_2$ emissions, based on IEA 2023 data[19]. Therefore, any underestimation arising from the excluded targets is unlikely to materially affect our conclusions on China's energy transition and future carbon emissions, although assessing these effects more comprehensively is an important direction for future research.

In contrast to existing studies of China's climate and energy policies, our scenario analysis quantifies the global influence of China's domestic policy credibility by explicitly linking it to global emissions and warming outcomes[20,21]. In doing so, we bridge the gap between policy credibility analysis and energy system modelling. This interdisciplinary perspective is essential for understanding the real-world feasibility of national climate pledges and their influence on national and global outcomes.

Our findings offer two key insights for global climate cooperation. First, assessing the credibility of climate policy is crucial for understanding both the boundaries of implementation uncertainty and its underlying drivers. Systematic, country-specific credibility assessments are essential for tracking national climate action, maintaining international trust, and fostering effective climate cooperation. Second, as a major economy, China has the capacity to compensate for insufficient action elsewhere in the medium term (Fig. 2c). Beyond direct emissions impacts, it is also in the broader interest of the rest of the world for Chinese climate policy to succeed, as continued leadership in technological innovation, manufacturing capacity, and cost reductions for clean technologies could accelerate global decarbonisation efforts. However, achieving the long-term goals of the Paris Agreement ultimately depends on sustained and collective global efforts.

## Methods
### Credibility assessment of Chinese climate policy delivery
Existing Chinese policy-related scenario studies have primarily focused on overarching goals such as the NDC, carbon peaking, and carbon neutrality targets[11,12]. These studies often assume the full and timely delivery of policy measures, while overlooking the uncertainties associated with delayed, partial, or failed implementation, particularly at the sectoral level. This paper takes a step toward closing that research gap by systematically assessing the credibility of China's sectoral climate and energy targets, providing a more realistic basis for evaluating whether the country is on track to meet its decarbonisation goals.

The definition of 'policy credibility' varies in the literature[22,23]. Furthermore, there is no standard assessment framework for policy credibility across studies[7,24]. Fransen et al. rely on cross-country composite indices such as the Climate Policy Score and the Climate Change Performance Index to capture policy adoption and outcome gaps[25]. Peterson and van Asselt propose five indicators to assess country-level NDC implementation risks: NDC ambition, institutional capacity, interest group opposition, policy inconsistency, and monitoring and enforcement, highlighting common challenges in closing the implementation gap[26]. While these frameworks are valuable for cross-country comparison, they are less suited to China's context, where sectoral targets embedded in central plans are the primary drivers of

emissions outcomes. In this study, we build on Rogelj, J. et al. who define credibility as "the level of confidence in a target delivery", and extend their net-zero target credibility assessment to national-level targets in China[6].

The policies reviewed in this study encompass all climate- and energy-related documents issued by the Chinese central government between 2020 and 2025, including action plans, implementation schemes, strategic outlines, guiding opinions, technical guidelines, and standards. In total, 58 policy documents were analysed, covering 292 numerical targets, including China's updated 2035 NDC.

Given the quantification limits of the TIAM-UCL model and the fact that some targets are unmeasurable (e.g., all ground vehicles and equipment at civil airports could strive to be powered by electricity by 2030), 47 targets were filtered and included in our credibility assessment (See Fig. 3b). The detailed reasons and implications for target non-quantifiability are discussed in SI Section 1. Target-level details are provided in the supplementary data. In the context of China, we developed and applied a credibility rating based on three key policy characteristics:

- **Governance level at which policy is set:** We replace the "legally binding" criterion in Rogelj et al. with the governance level at which a policy is set, reflecting China's hierarchical governance system, which is driven more by executive authority than parliamentary law[27,28]. Climate policies in China are typically issued as soft regulations rather than hard laws, making the level of authority rather than legal enforceability a more relevant indicator of credibility[29] (see Fig. 3a). For example, the "dual-carbon" targets (peaking carbon emissions by 2030 and achieving carbon neutrality by 2060), endorsed by the Chinese Communist Party (CCP) and State Council, carry the highest authority and are made binding at the national level. In contrast, policies from the National Development and Reform Commission (NDRC) or ministries are often non-binding guidelines. Higher-level policies tend to receive stronger institutional support, funding, and monitoring, making them more likely to be implemented effectively[30]. Lower-level policies, by contrast, rely heavily on local interpretation and capacity, often leading to inconsistent outcomes[31]. Thus, we treat targets with high-level endorsement and repeated top-down reaffirmation as more credible.

- **Clearly stated in the 14th Five-Year Plan (FYP):** The 14th Five-Year Plan (FYP) plays a key role in China's climate governance by integrating environmental goals into national development priorities and setting clear, strategic targets[32,33]. Moreover, the targets set in the 14th FYP are typically cautious and realistic, closely tied to government performance evaluations. This connection enhances their executive nature, making them more actionable and subject to evaluation at the end of the five-year period[2]. Such evaluations provide mechanisms for accountability and progress monitoring. Therefore, we assume that climate targets clearly stated in a Five-Year Plan are more actionable and credible due to their integration into performance evaluations, alignment with broader development priorities, and pragmatic approach to implementation.

- **Time normalised progress index (TNPI):** With China's 14th Five-Year Plan nearing completion, evaluating the extent to which each target is on track to be achieved could provide a foundation for developing new policies, plans, and targets for the next five years. Given the uncertainties in tracking broad goals like carbon neutrality and peak emissions, as well as the need for further model quantification, our assessment focuses exclusively on quantifiable, trackable, and statistically measurable targets (see SI Table 3). All evidence sources on progress are provided in the supplementary data.

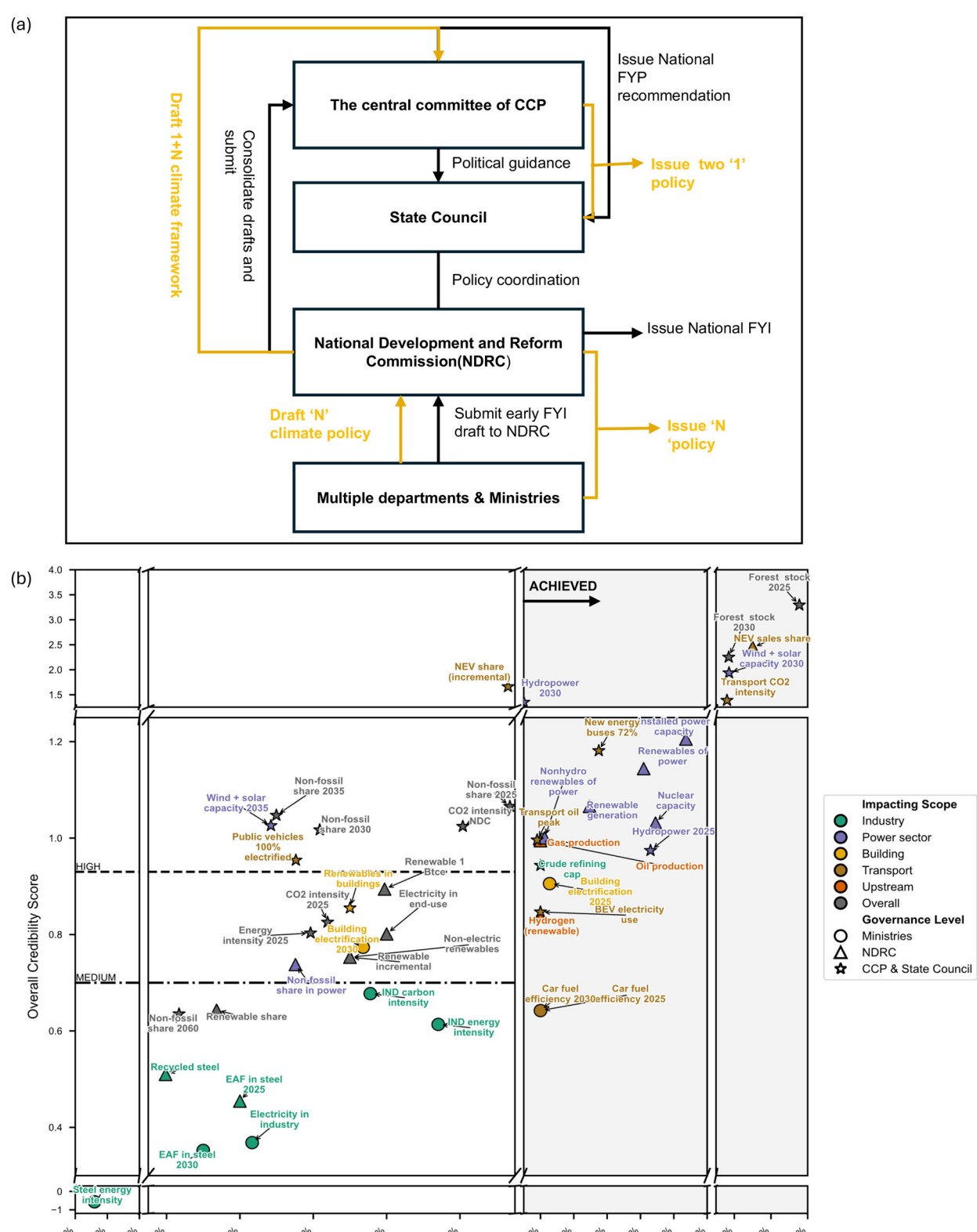

For each target, we first review and calculate raw progress as the proportion of the indicator gap that has been closed by the current year relative to the base year:

$$Raw\ progress = \frac{C_v - B_v}{T_v - B_v} \qquad (1)$$

Where $B_v$, $T_v$, $C_v$ are the base, target, and current values of the indicator. Targets where raw progress ≥100% are classified as Achieved. To ensure comparability across targets with different deadlines, we assume policy indicators advance along a linear trajectory and normalise progress by the proportion of time elapsed. This allows both short- and long-term goals to be evaluated based on whether progress is keeping pace with time elapsed. The resulting time-normalised

**Fig. 3 | China's climate governance architecture and cross-sectoral policy credibility assessment. a** Overview of China's climate governance framework led by the central government, adapted from Teng and Wang[29], Li et al.[48].;The black arrows indicate the formulation process of national Five-Year Plans, while the yellow arrows represent the development of the "1 + N" climate policy framework. Together, they illustrate the integration of Five-Year Planning and climate policy development. **b** Credibility assessment of China's climate and energy goals across sectors (details of each target assessment are provided in SI Section 2). Each shaped bubble represents a policy target. Bubble colour indicates the sector affected, and bubble shape reflects the level of government issuing the target (from the CCP and State Council to ministries, i.e., from higher to lower authority). The x-axis indicates the raw progress towards each target (as of October 2025), and the y-axis reports the final credibility score, calculated as a weighted average of the three evaluation criteria described in the text ($\omega_{gov} : \omega_{FYP} : \omega_{TNPI} = 3 : 2 : 5$). The grey region indicates targets that had been achieved or overachieved by the review date and are therefore treated as implemented from the GW scenario onwards. The dashed lines mark the credibility thresholds: high credibility (score ≥ the 67th percentile of all unachieved targets) and medium credibility (33rd–67th percentile).

progress index (TNPI) is given by:

$$TNPI = \frac{(C_v - B_v)/(T_v - B_v)}{(C_y - B_y)/(T_y - B_y)} * 100\% \qquad (2)$$

Where $B_y$, $T_y$, $C_y$ are the base, target, and current years. TNPI≥100% indicates the target is on or ahead of schedule; TNPI<100% indicates it is lagging. For infrastructure-related targets with long lead times (e.g., nuclear power, hydropower), we use operational and under-construction capacity as the current value to better account for construction lead times. All progress data are primarily based on official announcements and news releases from the Chinese government. When such information is unavailable, we make simple estimates using data from the IEA and public reports (e.g., Reuters). The specific data sources for each target's progress are listed in the supplementary data.

For targets still in progress (raw progress <1), we classify them using a weighted credibility score. Each target's score combines three components: governance level, inclusion in the 14th Five-Year Plan, and the TNPI. Figure 3b presents the overall credibility score using weights where $\omega_{gov} : \omega_{FYP} : \omega_{TNPI} = 3 : 2 : 5$. Targets are then clustered into categories as follows: High credible (score ≥ 67th percentile), Medium credible (33rd–67th percentile), and Low credible (below 33rd percentile). To evaluate the sensitivity of credibility scores to the weighting scheme and ensure the robustness of the clustering, we tested five alternative weighting schemes (see SI Section 1). Across weighting schemes, target classifications are largely robust. Only two targets shift category when the TNPI weight is lower than the other two indicators (see SI Table 13). We therefore adopt the consistent clustering results from the first four weighting schemes for our subsequent scenario quantification and report an additional weighting-related sensitivity case in SI Section 6.1. SI Figures 5 and 6 indicate that weighting-related sensitivity does not undermine our findings on China's energy transition and its emission implications.

In China's updated NDC, the targets for non-fossil energy share and installed wind and solar capacity extend existing pledges and are incorporated into our policy credibility assessment. However, the new GHG emissions target is China's first absolute reduction pledge, with no historical benchmark for evaluation; It is intended to be based on the peak year; however, since it remains unclear whether emissions have already peaked, we treat it as a policy sensitivity case (see SI Section 6.2).

## Scenario development with a morphological approach

In addition to lacking an exploratory perspective on policy credibility, many scenario studies of China adopt uniform socioeconomic assumptions drawn from global Shared Socioeconomic Pathways (SSPs)[11,34], or conduct additional sensitivity analyses only post hoc[12]. Such practices implicitly assume that socioeconomic development evolves independently of climate policy, neglecting the Chinese government's pivotal role in orchestrating economic transformation while managing trade-offs with its decarbonisation agenda.

The difference in climate policy implementation we assess reflects, through bespoke scenario narratives, the complex interactions among economic priorities, technological capabilities, governance capacity, and public engagement. Capturing this complexity requires a more flexible and context-specific approach to scenario design. Morphological analysis is the method used to visualise and analyse the complex network of relationships across the socio-technical system[35] (See Fig. 4a). A key advantage of this method is its transparency in combining multiple system drivers into a limited number of internally coherent scenarios. Morphological analysis is particularly well-suited to this, enabling the assessment of real-world decarbonisation pathways under uncertainty regarding governance and policy implementation in the Chinese context.

Our scenario design draws directly from official policy discourse and is anchored in China's current development challenges and structural constraints (SI Table 4). In response to mounting pressures from economic slowdown, the Chinese government has proposed a suite of high-level strategies to shape future national development. The resulting internally consistent scenarios serve two purposes: first, as plausible pathways for China's coordinated evolution across economic, demographic, technological, and policy dimensions; and second, as a structured exploration of how existing development strategies might interact under current socio-economic conditions to guide future trajectories. Together, these narratives offer a more context-specific framing for scenario modelling in China, bridging real-world policy signals with energy system analysis. For this study, four China-specific scenarios were developed. Here, we present a summary of each scenario. A comparative overview of key metrics across all four scenarios is reported in Fig. 4b. For more details on the scenario development process and narratives, including the national strategies referenced, please see SI Sections 2 and 3.

**Green lights (GL).** Green Lights depicts a future where China achieves high economic growth and climate leadership by 2060, reaching net-zero emissions by 2050. Through strategies like Dual Circulation[36] and the Beautiful China Initiative[37], the economy transitions to green, high-tech industries. Strong government action and falling technology costs drive electrification, innovation, and widespread public engagement. Clean tech deployment and carbon removal technologies, such as CCS and DAC, scale rapidly. Sustainable lifestyles gain traction, though fertility rates continue to decline. GL illustrates how climate ambition and economic prosperity can align, positioning China as a global model for low-carbon development.

**Calm sea (CS).** Calm Sea presents a balanced pathway where China maintains moderate economic growth while fully implementing current climate policies. The "new three" industries—solar, EVs, and batteries—drive decarbonisation and economic stability. China peaks emissions by 2030 and achieves carbon neutrality by 2060 through steady progress. Clean energy costs decline, enabling broad adoption, while CCS and DAC technologies scale moderately. Public behavioural change is policy-driven rather than voluntary. Economic reforms like Dual Circulation help avoid deeper downturns, but the lack of bold policies limits transformative change. CS reflects a pragmatic path, with policy stability supporting gradual decarbonisation.

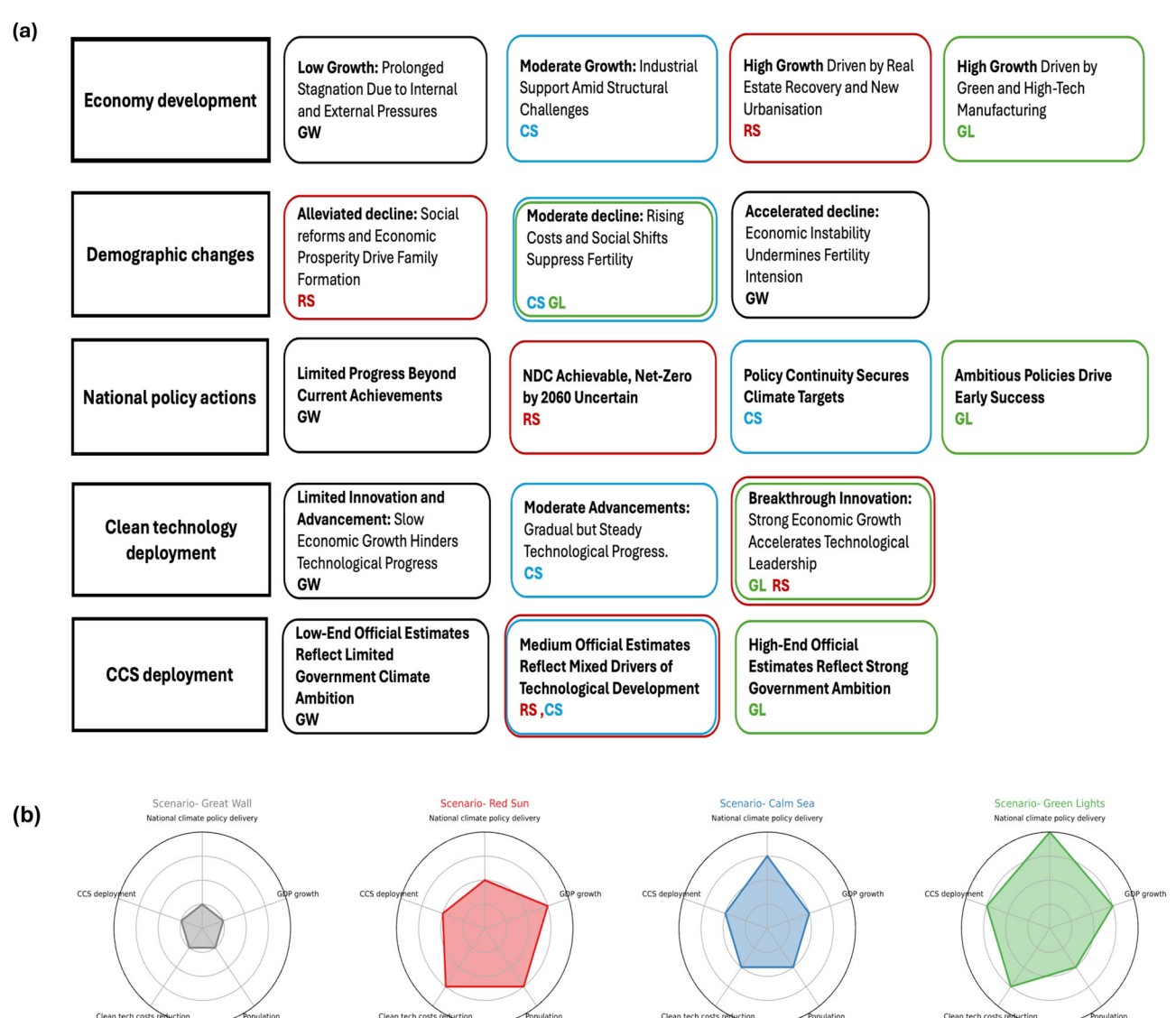

**Fig. 4 | Scenario framework and quantitative representation of key drivers in TIAM-UCL. a** Summary of morphological table, five key drivers are reviewed: economic development, demographic changes, national policy actions, clean technology development, and CCS deployment. Further details on the morphological analysis are provided in SI Table 4, and **b** Scenario quantification metrics. To quantify the scenario narratives in TIAM-UCL, we examine five key metrics and their associated uncertainties in SI Section 4, along with the corresponding modelling assumptions: national climate policy delivery (SI Section 4.1, Tables 5 and 6), GDP growth (SI Section 4.2, Fig. 3), population(SI Section 4.3, Tables 7 and 8), clean technology cost reductions(SI Section 4.4, Table 9), and CCS deployment rates(SI Section 4.5, Table 10). The degree of national climate policy delivery from low to high reflects the implementation of policies from 'achieved' to 'low credibility' levels. Numerical assumptions for the corresponding quantification metrics are detailed in SI Section 4.

**Red sun (RS).** Red Sun envisions a vibrant economy driven by real estate recovery[38] and urbanisation[39], with climate goals taking a back seat. China meets its 2030 NDC but struggles to reach net zero by 2060. Policy support focuses on economic growth, while industrial and carbon removal transitions are limited. Fertility rates stabilise due to social reforms and rising confidence. Consumerism and comfort take priority, with green tech adopted more for convenience than sustainability. Though emissions fall slightly through incremental technology uptake, weak climate governance and limited public engagement prevent China from making transformative progress.

**Great wall (GW).** Great Wall reflects a constrained future marked by economic stagnation, declining population, and stalled climate progress. High debt, a sluggish real estate sector, and weak consumption hinder economic reform. Technological innovation slows down, and clean energy deployment plateaus. CCS and DAC remain in early demonstration stages, and pro-natalist policies fail to halt population decline. Rising economic insecurity limits the government's capacity for sustained climate actions. China falls short of its 2025 climate targets and struggles to meet its 2030 NDC and 2060 net-zero goals. GW underscores the risks of limited government action and delayed response to climate and development challenges.

**Policy quantification across Chinese energy transition scenarios**
Policy quantification is guided by the assessed credibility of individual targets—classified as Achieved, High, Medium, or Low; and by scenario narratives developed using a morphological approach. The Great Wall scenario, which enforces only Achieved targets, therefore represents the most conservative pathway, reflecting minimal policy delivery. For targets that have already been achieved or overachieved, we use the realised levels observed by 2025 rather than the stated target values to more accurately capture China's existing climate efforts. Beyond the target year, these achieved levels are assumed to be maintained and form a minimum baseline from 2025 onwards.

**Table 1 | Framework of global climate scenarios incorporating uncertainties specific to China**

| The rest of the world (socio-economic assumption) | The rest of the world (climate actions) | China scenario |
|---|---|---|
| SSP 2 | NDC | Great Wall |
| SSP 2 | NDC | Red Sun |
| SSP 2 | NDC | Calm Sea |
| SSP 2 | NDC | Green Lights |
| SSP 2 | NDC + LTS | Great Wall |
| SSP 2 | NDC + LTS | Red Sun |
| SSP 2 | NDC + LTS | Calm Sea |
| SSP 2 | NDC + LTS | Green Lights |

Across scenarios, policy implementation expands progressively from higher- to lower-credibility targets. Moving from Great Wall to Green Lights, the assumed strength of policy delivery increases, with an expanding set of targets implemented, ultimately culminating in the full realisation of China's overarching climate commitment: carbon neutrality by 2060 or earlier (assumed as 2050). Detailed mappings between specific targets and scenarios are provided in SI Table 6.

### Global climate scenarios integrating Chinese uncertainty

To explore China's role in global climate mitigation, we model the above four China-specific scenarios under two potential levels of global climate action for the Rest of the World: Nationally Determined Contributions (NDC) and Long-Term Strategies (LTS) (See Table 1). Based on this approach, we can understand the implications of Chinese policy action in two different global contexts, one where the international community achieves limited ambition, based on NDCs, and another where stronger action is achieved, consistent with a 2 °C target. Although in reality China's climate actions would directly and indirectly influence other regions' ambition levels and global responses are unlikely to be entirely independent, this study focuses on isolating the global boundary effects of China's uncertainty. Therefore, we do not model dependencies or geopolitical interactions between China and the rest of the world.

First, an NDC case was developed based on Meinshausen et al.[40]. This study provides GHG emissions targets derived from NDCs for 2025 and 2030 for 196 countries as of November 2021, following the completion of COP26. It also includes emissions pledges for international aviation and shipping at the global level for these years. The NDC targets are aggregated to the TIAM-UCL regions, while international transport emissions (aviation and shipping) are distributed to the regions based on their 2020 shares of these emissions, according to IEA data.

To extend NDC ambition beyond 2030, we follow van de Ven et al., assuming each region's 2020–2030 emissions intensity reduction rate continues through the century[41]. To implement this, we use GDP projections from Shared Socioeconomic Pathway 2 (SSP2). Additionally, for this climate policy cases, we incorporate land-use, land-use change, and forestry (LULUCF) emissions and non-$CO_2$ emissions ($CH_4$ and $N_2O$) based on pathways already included in the TIAM-UCL model which draw on an average of four SSP2-RCP scenarios from the IMAGE, MESSAGE, REMIND and WITCH models[42]: the NDC case uses SSP2-RCP 6.0 scenario while the Net Zero case follows the SSP2-RCP 2.6 pathway. To better capture China-specific afforestation dynamics and reflect official forestation targets, we adopt China's LULUCF trajectory from He et al., which estimates the future carbon-removal potential consistent with national forestation targets[43].

Second, we represent more ambitious climate action in other regions by incorporating various net-zero pledges[41]. These pledges are applied to the regions listed in Table 2a, which covers countries and

**Table 2 | Long-term net-zero pledges and assumptions in TIAM-UCL (excl. China)**

| (a) Countries/regions with explicit GHG net-zero pledges | |
|---|---|
| **Region** | **Year** |
| Australia | 2050 |
| Canada | 2050 |
| Europe | 2050 |
| India | 2070 |
| Japan | 2050 |
| South Korea | 2050 |
| Mexico | 2050 |
| UK | 2050 |
| USA | 2050 |

| (b) Net-zero pledge coverage (GHG) rate and assumptions for TIAM selected regions | | | | |
|---|---|---|---|---|
| | **2050** | **2060** | **2070** | **2100** |
| Africa | 42.71% | 52.10% | 52.10% | 100% |
| Central and South America | 72.29% | 72.29% | 72.29% | 100% |
| Former Soviet Union | 8.37% | 86.55% | 86.55% | 100% |
| Middle East | 31.98% | 61.96% | 61.96% | 100% |
| Other Developing Asia | 49.78% | 81.35% | 90.50% | 100% |

regions with explicit net-zero targets. For regions where national commitments vary widely (Table 2b), we assume that each region partially achieves net zero based on the emission share of countries within it that have adopted net-zero targets, and all these regions converge to net-zero GHG emissions by 2100, even if some countries have not yet adopted net-zero targets. In the modelling, we aggregate country pledges and report the share of 2020 regional GHG emissions covered by net-zero commitments by 2050, 2060, and 2070, and implement this proportion as GHG emissions reduction relative to the NDC baseline case. This approach reflects the ambition of major emitters within each region while avoiding over-reliance on early pledges from smaller countries.

### The TIAM-UCL model

To explore the implementation of our China-focused policy scenarios and their global implications, we use the TIMES Integrated Assessment Model at University College London (TIAM-UCL). This model provides a representation of the global energy system, capturing primary energy sources (oil, fossil methane gas, coal, nuclear, biomass, and renewables) from production through to their conversion (electricity production, hydrogen and biofuel production, oil refining), their transport and distribution, and their eventual use to meet energy service demands across a range of economic sectors. Using a scenario-based approach, the evolution of the system over time to meet future energy service demands can be simulated, driven by a least-cost objective. The model uses the TIMES model framework.

The model represents the countries of the world as 16 regions, enabling detailed characterisation of energy systems in each region and the trade flows between regions. Regional coal, oil, and gas prices are generated within the model. These incorporate the marginal cost of production, scarcity rents (e.g., the benefit foregone by using a resource now as opposed to in the future, assuming discount rates), rents arising from other imposed constraints (e.g., depletion rates), and transportation costs, but not fiscal regimes. This means full price formation, which includes taxes and subsidies, is not captured in TIAM-UCL, and remains a contested limitation of this type of model[44].

The model has a limited number of technological options for carbon dioxide removal (CDR) from the atmosphere, including a set of

bioenergy with carbon capture and storage (BECCS) technologies, in power generation, industry, and in hydrogen and biofuel production. The climate module used in the model enables the translation of greenhouse gas emissions from the energy system into atmospheric concentrations, radiative forcing, and temperature change. The module is mainly based on that developed by ETSAP[45], but has been subsequently recalibrated to the MAGICC model results[46]. Further details on recent model developments can be found in Pye et al.[47].

Future demands for energy services (including mobility, lighting, residential, commercial and industrial heat and cooling) are exogenously defined and drive the evolution of the system so that energy supply meets demand throughout the time horizon. Projections of China's energy service demand are based on GDP and population assumptions specific to each scenario (SI Figure. 5), aligning with narratives of China's economic transition (SI Section 4). Energy service demands for the rest of the world are derived from SSP2. Decisions around what energy sector investments to make across regions are determined based on the cost-effectiveness of investments, taking into account the existing system today, energy resource potential, technology availability, and policy constraints.

TIAM-UCL runs in 5-year steps from 2005 to 2100, calibrated to IEA data and updated for 2010–2020 with additional constraints introduced to represent the existing system today. The social discount rate used in the net present value calculation (the basis for the objective function) is set at 3.5%. Further information on key assumptions used in the model is provided in SI section 4. The TIAM-UCL model version used for this analysis was 4.1.2, and the analysis was run using TIMES code 4.5.6 with GAMS 38. The model solver used was CPLEX 20.1.0.1.

## Reporting summary
Further information on research design is available in the Nature Portfolio Reporting Summary linked to this article.

## Data availability
The processed data underlying the figures and analyses presented in this study, including source data for the policy credibility assessment, net-zero assumptions, and figure generation, have been deposited in the Zenodo repository (https://doi.org/10.5281/zenodo.18378439). The TIAM-UCL model outputs and files with the assumptions needed to run the model are available in the same repository.

## Code availability
The underlying code (mathematical equations) for the model is available via GitHub (https://github.com/etsap-TIMES/TIMES_model).

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

## Acknowledgements

S.P. and J.P. involvement was supported by the Horizon Europe R&I programme project DIAMOND (grant no. 101081179).

## Author contributions

D.Z., S.P., J.P., and J.W. designed the study, with contributions from D.W. D.Z. and J.P. conducted the TIAM-UCL modelling, with contributions from D.W. D.Z. led the presentation of the modelling results. D.Z. and S.P. led on the drafting of the manuscript, with contributions from all other authors. D.Z. led the compilation of the supplementary information.

## Competing interests

The authors declare no competing interests.
