## [Transparent Peer Review file · Nature Communications]

Global implications of uncertainty in China's climate policy delivery

Corresponding Author: Ms Dan Zhang

Version 0:

Reviewer comments:

Reviewer #1

(Remarks to the Author)

This study combines an energy system optimization model with a policy credibility-based scenario framework to evaluate the credibility of China's climate policy targets and assess the implications of different credible policy packages for future emissions pathways and cumulative emissions in China and globally. I find the overall research design to be clever and the use of the UCL-TIAM model appropriate and well-established in the literature. However, I believe there is a fundamental and potentially unresolvable issue with the policy credibility metric that significantly undermines the validity of the main conclusions. Below I explain why this issue is critical and how it affects the interpretation of the results.

In the Methods section (Lines 329–330), the authors construct a composite policy credibility indicator comprising three dimensions: (1) administrative level of the policy target (higher levels score higher), (2) whether the target is included in China's Five-Year Plan (included targets score higher), and (3) the level of progress toward the target. The third dimension is quantified as the ratio of current progress to expected progress, i.e., $(\text{current} - \text{base}) / (\text{target} - \text{base})$. A lower ratio indicates lower credibility.

This metric, however, suffers from two critical flaws:

First, it assesses indicator progress, not policy progress. In practice, indicators often lag behind policy implementation, especially for infrastructure-related targets. For example, the authors assess the credibility of China's nuclear power target as very low (see Methods Figure b and SI), based on the gap between current installed capacity in 2023 and the 70 GW target for 2025. However, due to the long lead time of nuclear construction (6–7 years), many projects initiated during the 14th Five-Year Plan will only come online by 2025, and their progress is essentially certain. A more appropriate indicator would be installed + under-construction capacity or projected 2025 operational capacity. Using these metrics would yield significantly higher credibility scores for the nuclear target.

Second, the indicator is highly sensitive to the time horizon of the target. For instance, the authors assess China's non-fossil energy share targets for 2025, 2030, and 2060. Because the numerator (current – base) is constant across targets, longer-term targets with larger denominators result in lower credibility scores. This artificially favors short-term targets as more “credible” and penalizes long-term ones. As a result, the scenario grouping by credibility is effectively a grouping by time horizon. The conclusions are thus driven by whether China meets its long-term targets, not whether the targets are credible per se. This weakens both the policy relevance and academic contribution of the study.

I am not sure whether this issue can be resolved without revisiting the scenario design, but I believe it has serious implications that challenge the main conclusions of the paper.

In addition, I offer the following specific comments:

1. The paper does not include scenarios or discussion on carbon sinks, which are an essential component of China's carbon neutrality target but not of the peaking target. The credibility assessment should include carbon sink-related targets where applicable.
2. In Methods Figure b, I recommend using different colors to distinguish policy targets by sector to improve readability.
3. In Figure 2, panels a and b end at 2060, while panels d and f report cumulative emissions and warming until 2100. The emissions pathways in a and b should be extended to 2100 for consistency.

4. Since official targets only extend to 2060, all post-2060 assumptions (such as maintaining emissions decline rates) are hypothetical. These assumptions appear to be the dominant driver of the differences in cumulative emissions by 2100. For example, in Figure 3c, the difference in cumulative emissions by 2050 is less than 100 GtCO₂, whereas the difference from 2050 to 2100 exceeds 400 GtCO₂. The authors' claim in the abstract that "uncertainty in the delivery of China's domestic policy alone could lead to a difference in cumulative global emission reductions of up to ~600 GtCO₂ by 2100, equivalent to a ~0.2 °C warming difference" appears to significantly overstate the role of policy credibility.
5. In Figure 2b, the GW and RS scenarios suggest China's power sector is nearly decarbonized by 2060. Yet Figure 2c shows continued coal consumption in the power sector under those same scenarios (purple bar). This inconsistency needs to be clarified.
6. Figure 2 indicates that inter-scenario differences are largely driven by the extent of industrial electrification and coal reduction, rather than differences in the power sector. However, the conclusion emphasizes coal phaseout in power generation as the key. This is not well supported by the results.
7. The Discussion section lacks close integration with the results. The three key messages listed in the discussion are not clearly linked to the earlier findings and should be revised for consistency.
8. The descriptions of the four scenarios are only found in the Methods section. A brief summary of each scenario in the main text would greatly improve accessibility and comprehension for readers.

Reviewer #2

(Remarks to the Author)

This review addresses the manuscript titled "The global implications of uncertainty in China's climate policy delivery," submitted to Nature Communications. The paper aims to quantify the effect of uncertainty in China's climate policy delivery on global emissions and temperature outcomes. It addresses the following questions: (1) How credible are China's climate and energy policies? (i.e. how likely is it that the targets they set will be achieved?) (2) Taking into account the uncertainty stemming from varying credibility, what are the range of emissions outcomes for China over the course of the century? (3) What are the implications of this range for global temperature outcomes under two different scenarios for global ambition?

The paper is both innovative and policy relevant. It makes three main contributions. First, it adapts a generic framework for assessing policy credibility to the specific institutional and political context of China, providing a methodological advance. Second, it develops a novel dataset that transparently evaluates the credibility of China's climate and energy targets. Third, it quantifies the substantial implications of uncertainty in China's policy delivery for both emissions trajectories and global temperature outcomes.

Addressing several issues would further strengthen the paper:

A first issue concerns the scope of targets assessed. The paper states that 289 total targets were identified, of which 87 were quantifiable in TIAM, but fewer than 50 were ultimately assessed for credibility. It would be helpful to clarify the difference between the 87 and the <50 – is the difference due entirely to local/regional targets? More critically, how do the excluded targets factor into the scenarios, if at all? Are they assumed to have no impact, and if so, is that assumption defensible? Since the excluded targets are unlikely to be random (for instance, energy-sector targets are more readily quantifiable in TIAM, whereas land-use or efficiency targets may be less so), the criteria for inclusion or exclusion could introduce bias. It would be helpful to clarify the rationale for this filtering step and discuss how it might affect the results.

A second issue relates to the credibility assessment methodology. The framework and the choice of indicators (governance level, inclusion in the Five-Year Plan, and progress to date) are logical. There is a missed opportunity, however, to engage with literature discussing factors in the target implementation gap (e.g. 1, 2), which could provide a stronger foundation for the indicators chosen. Additionally, while the rationale for weighting progress to date higher than the other two indicators is clear, the weightings in general seem somewhat arbitrary. How sensitive are the results to the weightings chosen?

A third issue concerns the treatment of targets after their stated target year. While the paper is clear about which targets are assumed to be achieved in which scenarios, it is less clear how this achievement is translated into the modeling framework, and in particular what was assumed about progress on the target indicators following the target year. There are two potentially conflicting logics: On the one hand, one could interpret an "achieved" target as evidence of strong institutional capacity and assume continued progress thereafter. Alternatively, an "achieved" target could reflect an unambitious goal, which might correlate with weak implementation capacity, in which case it would not signal a trajectory of sustained improvement. I was not able to decipher how such logic is carried forward from the credibility assessment into the scenario narratives and ultimately into the IAM results. Greater clarity on this point would be valuable.

A fourth set of issues relates to the global net zero scenario. The accuracy of assumptions in Table 2 should be confirmed. For instance, while emission coverage for Central and South America is said to cover CO₂ only, the net-zero targets of the largest countries in the region (e.g. Brazil, Argentina, Chile, Peru, Colombia) cover all GHGs (3, 4). Likewise for the former Soviet Union. The basis for choosing a "representative" country for certain regions should also be clarified – are these countries representative in terms of size of economy, emissions profile, or some other indicator? (For example, on what basis is Nigeria said to be representative of Africa?) Alternatively, it is plausible that the results are not particularly sensitive to these assumptions, in which case, it would be helpful to clarify that.

References

1. L. Peterson, H. and van Asselt, Assessing risks to the implementation of NDCs under the Paris Agreement. Clim. Policy 0,

1–15.

2. T. Fransen, J. Meckling, A. Stünzi, T. S. Schmidt, F. Egli, N. Schmid, C. Beaton, Taking stock of the implementation gap in climate policy. *Nat. Clim. Change*, 1–4 (2023).

3. Climate Watch, Net-Zero Tracker (2024). <https://www.climatewatchdata.org/net-zero-tracker>.

4. Energy and Climate Intelligence Unit, Data Driven Envirolab, New Climate Institute, University of Oxford, Net Zero Tracker, (2024); <https://zerotracker.net/>.

Version 1:

Reviewer comments:

Reviewer #1

(Remarks to the Author)

In this revised version, the authors have substantially strengthened the manuscript and have addressed most of the concerns raised in the previous review. The paper now presents a timely and policy-relevant assessment of China's climate policy credibility and its global implications, combining a structured policy review with scenario-based integrated assessment modelling. I have only a few remaining minor points—mainly clarifications for transparency and small figure/presentation refinements—that would further improve readability and reproducibility before publication.

1. The manuscript/abstract could be read as implying that the structured credibility assessment was applied to all identified targets (e.g., 292 targets across policy documents). However, the Supplementary Information indicates a multi-step filtering process: 292 numerical targets were identified, then a subset of “quantifiable” targets was retained (reported as 90, including duplicates), which was further consolidated to a final set of 47 targets that were actually evaluated/scored for credibility. Please clarify this explicitly in the main text/abstract to avoid confusion—e.g., state that the study identifies 292 targets but the credibility scoring is conducted on the filtered set of quantifiable targets (final N=47), and briefly explain the filtering rationale

2. While the manuscript states that scenario narratives are informed by the policy credibility assessment, the mapping between individual policy credibility scores and the quantified scenario assumptions remains insufficiently specified. This may lead to confusion for non-specialist readers regarding the purpose of the credibility assessment, how it influences scenario construction, and how it ultimately affects the modeling results. Given that relevant tables and detailed mappings already appear to be provided in the Supplementary Information, the authors are encouraged to add a brief explanatory bridge in the Introduction and/or Methods sections.

3. The authors are encouraged to carefully check the consistency and completeness of cross-referencing between the main text and the Supplementary Information (SI). At present, several tables and figures in the SI appear not to be explicitly cited in either the main text or within the SI itself. For example, in the SI (around line 140), the text refers to “(see SI Table X, Step 3)”, the corresponding table reference is unclear. In addition, SI Tables 4, 5, 7, 8, 11, and 12 do not appear to be cited anywhere in the main manuscript or the SI text. A similar issue applies to several supplementary figures.

4. Figure 1(b): The dashed line indicating the share of renewable energy (black) overlaps with the coal bar for the year 2020, which is also shown in black. This makes it difficult to identify the renewable share value for that year.

5. Figures 2(a) and 2(b): In the GW scenario, the grey uncertainty band overlaps with the red shaded area representing the RS scenario, resulting in a bluish color that is visually ambiguous. Adjusting the plotting order (layering) or modifying transparency/color choices would help avoid color blending and improve interpretability.

6. Figure 3(a): Some text elements are obscured or truncated. On the left-hand side, labels are partially covered by lines, while on the right-hand side the word “recommendation” appears broken or cut off. Please revise the layout to ensure all text is fully visible and legible.

7. Figure 3(b): (1) The governance level indicator (from the CCP to ministries) is currently encoded solely through bubble size. However, the visual contrast between sizes appears insufficient to clearly distinguish different administrative levels, particularly between the largest bubbles (CCP & State Council) and the medium-sized bubbles (e.g. NDRC). The authors may consider combining symbol shape with size to enhance visual differentiation—for example, using large circles for CCP-level targets and smaller symbols (e.g. triangles or stars) for targets issued by the NDRC and ministries.

(2) The figure contains a large number of data points and employs multiple discontinuous or non-linear axis segments, which results in substantial crowding in key regions (e.g., credibility scores between ~0.7 and 1.1). This leads to overlapping labels and makes it difficult to identify individual policy targets. The authors may wish to explore alternative visualization strategies, such as segmented or logarithmic scaling, selective zoom-in in critical ranges, or inset panels focusing on dense regions, to improve legibility.

(3) In light of the scenario narratives developed in this section, the figure could be further strengthened by explicitly highlighting or grouping key clusters of targets. For example, distinguishing targets with high credibility scores but lagging progress (which may require strong policy reinforcement as represented in the GL scenario), versus targets with low credibility yet high mitigation relevance (which may be deprioritized under the GW scenario), would enhance the narrative coherence between the credibility assessment and the subsequent scenario analysis.

8. Line 356: The text states “Where B_v, T_v, T_v are the base, target, and current years”, but this should likely write

B_y,T_y,T_y.

9. Line 365-366: While the manuscript states that the credibility score combines three components and tests five alternative weighting schemes (SI Section 1), the main text does not specify the actual weights applied to each indicator, nor clarify which weighting scheme(s) underpin the results shown in Fig. 3 and the subsequent scenario analysis. Given that the credibility score plays a central role in informing scenario narratives, a brief description of the weighting logic and reference to the relevant SI tables in the main text would improve transparency and reproducibility.

10. The analysis focuses on 47 quantifiable targets due to model constraints and data availability. While this filtering is understandable, it may introduce a systematic selection bias by excluding non-quantifiable but policy-relevant measures (e.g., water, waste, recycling, air pollution control, and detailed industrial or building efficiency policies). As a result, mitigation contributions from easily measurable sectors such as power and transport may be overstated, which may shift sectoral emphasis and alter perceived feasibility/costs; please discuss likely direction and (if possible) sensitivity. The authors are encouraged to discuss how this sectoral imbalance may affect the estimated feasibility, costs, and structural characteristics of China's net-zero transition, and to clarify the likely direction of the bias introduced.

11. The first Results subsection ("Chinese Decarbonization and Energy Transformation") presents outcomes of China's energy system transformation under different domestic policy scenarios (GW–GL) without explicitly stating the assumed global context. Please specify which global context is used for Fig. 1, and briefly note whether the qualitative conclusions are robust across both contexts. Nevertheless, a brief clarification in the text—stating that these results are "global context–neutral" or not materially affected by ROW assumptions—would help avoid potential confusion for readers who might otherwise expect Figure 1 to correspond to a specific global scenario.

Reviewer #2

(Remarks to the Author)

I reviewed an earlier version of this manuscript and found it to be a strong contribution, pending revisions. I have now reviewed the revised version and the authors' responses to reviewers. I am satisfied that the authors have adequately addressed the concerns raised in the initial round of review. Their revisions are appropriate and improve the clarity and robustness of the manuscript. I recommend the manuscript for publication.

Version 2:

Reviewer comments:

Reviewer #1

(Remarks to the Author)

The authors have carefully addressed and incorporated the comments from the previous review round and I recommend acceptance for publication

Responses to Reviewers

NCOMMS-25-41170-T

October 2025

Reviewer 1

General comments:

This study combines an energy system optimization model with a policy credibility-based scenario framework to evaluate the credibility of China's climate policy targets and assess the implications of different credible policy packages for future emissions pathways and cumulative emissions in China and globally. I find the overall research design to be clever and the use of the UCL-TIAM model appropriate and well-established in the literature. However, I believe there is a fundamental and potentially unresolvable issue with the policy credibility metric that significantly undermines the validity of the main conclusions. Below I explain why this issue is critical and how it affects the interpretation of the results.

In the Methods section (Lines 329–330), the authors construct a composite policy credibility indicator comprising three dimensions: (1) administrative level of the policy target (higher levels score higher), (2) whether the target is included in China's Five-Year Plan (included targets score higher), and (3) the level of progress toward the target. The third dimension is quantified as the ratio of current progress to expected progress, i.e., $(\text{current} - \text{base}) / (\text{target} - \text{base})$. A lower ratio indicates lower credibility.

This metric, however, suffers from two critical flaws:

First, it assesses indicator progress, not policy progress. In practice, indicators often lag behind policy implementation, especially for infrastructure-related targets. For example, the authors assess the credibility of China's nuclear power target as very low (see Methods Figure b and SI), based on the gap between current installed capacity in 2023 and the 70 GW target for 2025. However, due to the long lead time of nuclear construction (6–7 years), many projects initiated during the 14th Five-Year Plan will only come online by 2025, and their progress is essentially certain. A more appropriate indicator would be installed + under-construction capacity or projected 2025 operational capacity. Using these metrics would yield significantly higher credibility scores for the nuclear target.

Second, the indicator is highly sensitive to the time horizon of the target. For instance, the authors assess China's non-fossil energy share targets for 2025, 2030, and 2060. Because

the numerator (current – base) is constant across targets, longer-term targets with larger denominators result in lower credibility scores. This artificially favors short-term targets as more “credible” and penalizes long-term ones. As a result, the scenario grouping by credibility is effectively a grouping by time horizon. The conclusions are thus driven by whether China meets its long-term targets, not whether the targets are credible per se. This weakens both the policy relevance and academic contribution of the study.

I am not sure whether this issue can be resolved without revisiting the scenario design, but I believe it has serious implications that challenge the main conclusions of the paper.

Response:

Thank you for the constructive comments, which highlight two limitations in our initial credibility assessment approach. In this revision, we have updated the progress review of all targets involved, taking account power plant capacity under construction for relevant targets, and adjusted the progress indicator to capture the potential time horizon issues.

Our responses to this methodological issue are organised around the following three points:

1. Policy progress versus indicator progress

In this study, our objective is not to evaluate policy progress per se, but rather policy credibility. The progress index is one of the metrics used to assess whether a policy indicator is on track, while the other two dimensions, governance level and inclusion in the Five-Year Plan, reflect the authority and institutional strength of the policy. We acknowledge that some targets may show limited progress to date yet still be considered credible, particularly those with mid-term timelines. This is why we adopted a weighting approach in which governance levels assigns additional credibility to mid- or long-term targets. However, we recognise that our initial weighting method involved a degree of arbitrariness. Therefore, in this revision, we introduce five alternative weighting schemes, including one in which the progress indicator carries a lower weight relative to the other two. This new configuration generates a sensitivity case, which is discussed in detail in our response to Reviewer 2's comment 2.

2. Infrastructure-related indicator progress lag

Thank you for pointing out the issue of construction lead time in progress assessment. We agree that this construction lead time issue is particularly relevant for infrastructure/technology-related indicators, though it is less significant across other targets. Therefore, in this revision, we now account for both operational capacity and under-construction capacity of nuclear when evaluating such indicators, which better reflects policy implementation progress and delivery certainty. Similar adjustments have been applied to hydropower targets.

3. Sensitivity of credibility scores to time horizon

We acknowledge the reviewer's concern that our original metric may be sensitive to target time horizons, but we do not agree with the view that this introduces significant bias and challenges our conclusions. First, as noted earlier, each target's credibility score comprises three components. Long-term targets generally have higher governance levels and are therefore not artificially penalised. Second, our scenario allocation does not simply follow year target, that is nearer term targets are not necessarily more credible. For example, several 2030 targets are already implemented in Red Sun due to their high credibility (e.g., EV, solar, and wind targets), while some 2025 industrial targets are only realised in Green Lights because of their lower credibility. In this sense, targets are not grouped into scenarios by time horizon. Third, while we constructed four distinct China-specific scenarios to explore potential variations in its energy transition, our key conclusions on its global implications are derived from the differences between these scenarios. The critical boundary lies between Great Wall (which includes only achieved or overachieved targets) and Green Lights (which implements all policies issued). Thus, the main findings regarding China's policy delivery and its global implications remain robust.

However, we do recognise the benefits of improving comparability across targets with different deadlines and reducing uncertainty in policy assessment. To address this limitation, we made two key revisions:

- Updated progress review: The earlier version of the manuscript assessed target progress as of July 2024. We now provide a comprehensive update as of 28 September 2025, incorporating the latest government press releases, statistical bulletins, and industry reports. This ensures that short-term (2025) targets are evaluated against the most up-to-date evidence of implementation.
- Time-normalised progress index (TNPI): We adjust our raw progress index to time normalised progress index (TNPI). TNPI is calculated as $Raw\ progress / Time\ fraction$, where $Raw\ progress = (current - base) / (target - base)$, and $Time\ fraction = (current\ year - base\ year) / (target\ year - base\ year)$. TNPI compares the share of the target gap closed with the share of time elapsed. This adjusted indicator is clarified to assess if targets with different deadlines are on track. For example, while the raw progress toward the 2060 non-fossil share target is only 3.5% by 2023, the TNPI indicates 45.3% when normalised for time. However, this treatment includes an implicit assumption that policy indicators advance along a linear trajectory; we have clarified this assumption in the manuscript.

Together, these revisions strengthen the robustness of our credibility assessment. The relevant revisions are reflected in the manuscript (lines 339-376) and in the Supplementary Information (lines 71-159). All updated data sources, calculations, and the revised credibility scores are provided in the SI_PolicyCredibilityAssessment.xlsx

Comment 1.1

The paper does not include scenarios or discussion on carbon sinks, which are an essential component of China's carbon neutrality target but not of the peaking target. The credibility assessment should include carbon sink-related targets where applicable.

Response:

Thank you for highlighting the importance of carbon sinks in the context of China's carbon neutrality target. Both engineered and natural sinks are considered in our scenario framework; however, we recognise the limitations of our previous treatment and have made corresponding enhancements in this revision.

For engineered sinks (CCS/BECCS/DAC), while China currently lacks a quantified national deployment target, several demonstration and build-out plans exist. Accordingly, we did not include a quantified policy-based CCS target in the credibility assessment, but we do incorporate CCS deployment uncertainty as a scenario dimension across all scenarios. The CCS assumptions, derived from China's official CCUS roadmap and a systematic review of deployment potential, are detailed in the SI section 4.5.

For natural sinks (LULUCF), the TIAM-UCL model represents CO₂ emissions from land use, land-use change, and forestry at the regional level, using exogenous inputs from other integrated assessment models. In our NDC scenarios, we adopt the average LULUCF trajectory from four SSP2–RCP6 pathways (IMAGE, MESSAGE, REMIND, and WITCH). For the net-zero scenarios, we use a global LULUCF trajectory consistent with SSP2–RCP2.6 (IMAGE), which delivers net-negative CO₂ emissions from 2060 onward.

However, we recognise that this generic treatment does not fully capture China-specific afforestation dynamics and is not closely aligned with China's official forestation targets:

- By 2025, the forest coverage rate will reach 24.1%, and the forest stock volume 18 billion m³.
- By 2030, the forest coverage rate will reach about 25%, and the forest stock volume 19 billion m³.

Given the structural limits of TIAM's land-use sector, it is challenging to represent forest coverage or stock volume directly. In this revision, we therefore parameterise China's LULUCF trajectory based on He et al. (2024), who implemented OSCAR-China to estimate future carbon-removal potential consistent with national forestation targets. Their study shows that, if these targets are achieved and extended, China's land sinks could reach -0.35 ± 0.04 Gt C yr⁻¹ by 2060¹.

Our updated policy assessment indicates that China has already exceeded its near-term forestation goals. Accordingly, in the revised modelling, these targets are classified as "ACHIEVED" and are

implemented in the Great Wall scenario and across all of the other more ambitious scenario. Additional clarifications and corresponding revisions have been incorporated in the manuscript (lines 478-485).

Comment 1.2

In Methods Figure b, I recommend using different colours to distinguish policy targets by sector to improve readability.

Response:

Thank you for this helpful suggestion. In the revised manuscript, we have expanded the policy credibility assessment to include additional targets (e.g., forest stock and the 2035 NDC), with progress updated to the most recent evidence (September 2025). To improve readability, we now use different colours and text labels to distinguish policy targets by sector, following the same colour scheme as in the results figures. Given some of the targets that have been substantially overachieved (such as forest stock and wind/solar capacity), we have also adopted a broken-axis figure to better illustrate the large variation in progress across targets. This approach improves readability and more clearly reflects both China’s significant achievements and areas where progress lags. The revised figure is included in the manuscript.

Figure 3.b Credibility assessment of China’s climate and energy goals across various sectors (with details of each target assessment provided in SI section 1).

Comment 1.3

In Figure 2, panels a and b end at 2060, while panels d and f report cumulative emissions and warming until 2100. The emissions pathways in a and b should be extended to 2100 for consistency.

Response:

Thank you for this helpful suggestion. In the revised manuscript, we have extended the emissions pathways in Figure 2 panels a and b to 2100, ensuring consistency with panels d and f. The changes in the net-zero group for the rest of the world result from updated net-zero assumptions for key developing regions, as detailed in our response to Reviewer 2, Comment 2.4.

Comment 1.4

Since official targets only extend to 2060, all post-2060 assumptions (such as maintaining emissions decline rates) are hypothetical. These assumptions appear to be the dominant driver of the differences in cumulative emissions by 2100. For example, in Figure 3c, the difference in cumulative emissions by 2050 is less than 100 GtCO₂, whereas the difference from 2050 to 2100 exceeds 400 GtCO₂. The authors' claim in the abstract that "uncertainty in the delivery of China's domestic policy alone could lead to a difference in cumulative global emission reductions of up to ~600 GtCO₂ by 2100, equivalent to a ~0.2 °C warming difference" appears to significantly overstate the role of policy credibility.

Response:

Thank you for raising this concern regarding the influence of post-2060 assumptions on China's policy impact. We acknowledge that these assumptions inevitably affect the absolute values of cumulative emissions and temperature outcomes by 2100. However, our analysis focuses on the differences arising from variations in China's policy delivery and socio-economic–technological development (the other four dimensions in our morphological analysis), rather than on the absolute levels themselves.

This study evaluates policies issued up to 2025, excluding potential future policies that have not yet been announced, as policy projection lies beyond the scope of our research. Our analysis assumes that policy ambition is sustained, at minimum, at current levels of effort (Great Wall). Our findings therefore represent a scenario-based assessment rather than a prediction or forecast. The long-term outcomes up to 2100 reflect the implications of existing policies and foreseeable socio-economic conditions (long-term assumptions about economic, demographic, and technological trends are clarified in SI Section 4). This is precisely what we aim to evaluate. Meanwhile, the differences shown in Figure 3c, approximately 75 GtCO₂ by 2050 and 500 GtCO₂ by 2100 (new modelling results), remain robust under two distinct global contexts (NDC and Net Zero). We therefore consider our conclusions regarding China's policy delivery implications to be robust within our scenario and modelling framework.

The slight reduction compared with our previous analysis (from ~600 GtCO₂ to ~500 GtCO₂, and from 0.20 °C to 0.17 °C of warming difference) arises from three key updates:

- Updated policy progress: We reassessed all policy targets using the most recent statistical data (as of September 28, 2025; the previous assessment was based on data up to July 2024), which show that a greater number of targets have now been achieved. The new assessment also updates the planned values of overachieved targets with actual progress data. Together, these revisions enhance and better reflect the mitigation potential of the most conservative (Great Wall) scenario.
- Inclusion of quantified forestation targets: the incorporation of updated, quantified forestation targets further enhances the mitigation potential across all scenarios. As these targets are assessed as achieved, their inclusion provides additional representation of land-based carbon removal within the model.
- Inclusion of 2035 NDC targets of non-fossil share and wind and solar capacity instalment.

Comment 1.5

In Figure 2b, the GW and RS scenarios suggest China's power sector is nearly decarbonized by 2060. Yet Figure 2c shows continued coal consumption in the power sector under those same scenarios (purple bar). This inconsistency needs to be clarified.

Response:

Thank you for the comment. Figure 2b presents the breakdown of electricity generation sources, while Figure 2c shows coal consumption for power sector include both electricity and heat generation. In the TIAM model, the power sector includes both electricity-generation technologies and heat-producing technologies (Public heat generation plants, providing heat to local networks), which explains the distinction between these two figures. Further details on the power sector configuration can be found in the TIAM documentation, Section 4.1 Power Generation².

In this revision, we separated heat production from Power and represented it in light purple in Figure 2. Additional clarification has been added to the figure caption (lines 175-177).

Comment 1.6

Figure 2 indicates that inter-scenario differences are largely driven by the extent of industrial electrification and coal reduction, rather than differences in the power sector. However, the conclusion emphasizes coal phaseout in power generation as the key. This is not well supported by the results.

Response:

Thank you very much for this insightful comment. In this revision, we have updated our policy credibility assessment to capture the most recent progress on numerical targets and to include projected progress in nuclear and hydropower installed capacity. As a result, more targets are now categorised as achieved, reflecting the strengthened policy efforts in the GW and RS scenarios. The revised results show more clearly that existing policy efforts ensure high certainty in the electrification expansion and power sector transition, while also highlighting the key role of industrial electrification and coal reduction which also drives inter-scenario differences. Accordingly, we have re-analysed the modelling results and revised the Discussion section to better align our conclusions with these updated modelling findings.

Comment 1.7

The Discussion section lacks close integration with the results. The three key messages listed in the discussion are not clearly linked to the earlier findings and should be revised for consistency.

Response:

Thank you for this comment. As a study combining qualitative and quantitative approaches, the Discussion section is designed to go beyond the numerical modelling results and to summarise broader insights from our morphological scenario analysis (a systemic assessment of China's economy, society, technology, and policy context) as well as from our policy review and assessment, including elements not directly quantified in the modelling framework. Given the methodological changes in policy credibility assessment and the updated net-zero assumptions for the rest of the world, we reran our scenarios and revised the Discussion to integrate the new findings (See lines 230-269). We believe this revision enhances consistency with the quantitative results and qualitative insights.

Comment 1.8

The descriptions of the four scenarios are only found in the Methods section. A brief

summary of each scenario in the main text would greatly improve accessibility and comprehension for readers.

Response:

Thank you for your helpful suggestion. We have added a brief summary of the four scenarios in the main text to clarify their differences and improve the readability for readers.

Manuscript line 60-65:

“The scenarios capture a diverse set of futures for China’s climate governance, alongside corresponding shifts in socioeconomic and technological development (see Methods and SI section 2). Great Wall (GW) reflects a constrained future marked by economic stagnation, declining population, and stalled climate progress. Red Sun (RS) envisions strong economic growth, driven by real estate market recovery and urbanisation, while climate goals are deprioritised. Calm Sea (CS) presents a balanced pathway where China maintains moderate economic growth while fully implementing current climate policies. Green Lights (GL) depicts a future where China demonstrates climate leadership while maintaining high economic growth.”

Reviewer 2

General comments

This review addresses the manuscript titled “The global implications of uncertainty in China's climate policy delivery,” submitted to Nature Communications. The paper aims to quantify the effect of uncertainty in China's climate policy delivery on global emissions and temperature outcomes. It addresses the following questions: (1) How credible are China's climate and energy policies? (i.e. how likely is it that the targets they set will be achieved?) (2) Taking into account the uncertainty stemming from varying credibility, what are the range of emissions outcomes for China over the course of the century? (3) What are the implications of this range for global temperature outcomes under two different scenarios for global ambition?

The paper is both innovative and policy relevant. It makes three main contributions. First, it adapts a generic framework for assessing policy credibility to the specific institutional and political context of China, providing a methodological advance. Second, it develops a novel dataset that transparently evaluates the credibility of China's climate and energy targets. Third, it quantifies the substantial implications of uncertainty in China's policy delivery for both emissions trajectories and global temperature outcomes.

Response:

Thank you for your careful review of our manuscript and for recognising the innovation and policy relevance of our study. We appreciate the constructive issues raised regarding specific aspects of scenario sensitivity and the modelling framework. Specifically, the revisions made in response to your four concerns are outlined below.

Comment 2.1

A first issue concerns the scope of targets assessed. The paper states that 289 total targets were identified, of which 87 were quantifiable in TIAM, but fewer than 50 were ultimately assessed for credibility. It would be helpful to clarify the difference between the 87 and the <50 – is the difference due entirely to local/regional targets? More critically, how do the excluded targets factor into the scenarios, if at all? Are they assumed to have no impact, and if so, is that assumption defensible? Since the excluded targets are unlikely to be random (for instance, energy-sector targets are more readily quantifiable in TIAM, whereas land-use or efficiency targets may be less so), the criteria for inclusion or exclusion could introduce bias. It would be helpful to clarify the rationale for this filtering step and discuss how it might affect the results.

Response:

Thank you for raising this important point. In the revised manuscript, we have enhanced the filtering process for policy targets and provided a clearer explanation of the steps involved, as well as their potential implications (SI Section 1 step 2, lines 131-149).

In this updated assessment, we also incorporated China’s newly announced 2035 NDC targets. Because our review begins from policy documents, we recorded all numerical targets contained in each document to ensure full coverage. This approach also allowed us to track the policy coverage frequency of each target later. In total, we identified 292 (287+3) targets across 58 policies, which included some double counting.

At the first filtering stage, each target was classified according to whether it was quantifiable. In this revision, we have documented the exclusion reason for each unquantifiable target in the accompanying SI_PolicyCredibilityAssessment.xlsx. The main reasons for exclusion are summarised as follows:

SI Table 2: Rationale for filtering TIAM-quantifiable targets

Rationale for filtering TIAM-quantifiable targets	Summary of Reasons	Examples of Targets
1. Technology / Commodity Not Represented	TIAM models technologies in aggregate but cannot capture disaggregated forms, infrastructure length, or building floor area. Similarly, some industrial commodities, such as nonferrous metals, are represented in aggregate, preventing explicit representation of specific sectors like aluminium and cement.	 • By 2025, the national oil and gas pipeline network will expand to approximately 210,000 km. • By 2025, the energy consumption per unit of cement clinker will be reduced by 3.7%. • By 2025, carbon emissions in the electrolytic aluminium sector will decrease by 5%.
2. Water, Waste, Recycling, and Non-Energy Resource Flows	Water use, recycling, waste treatment, secondary materials, and industrial by-products are outside TIAM’s scope. The model does not track paper, metals, slag, gypsum, manure, straw, red mud, or appliance/vehicle recycling, which are actually reflected in policies	 • By 2025, reduce water use per RMB 10,000 of industrial added value by 16% compared to 2020. • By 2025, achieve a nationwide water reuse rate of approximately 94% in large-scale industrial sectors.
3. Pollution, Non-GHG Metrics	TIAM cannot represent non-GHG pollutants, this includes PM2.5, COD, NOx, VOCs.	 • By 2025, total emissions of COD, ammonia nitrogen, NOx, and VOCs will decrease by 8%, 8%, >10%, and >10% respectively compared to 2020. • By 2025, operational ships will achieve a 7% reduction in NOx emissions compared to 2020 levels.

4. Industrial Standards & Economic / Financial Indicators	TIAM does not model regulatory compliance, benchmark standards, retrofits, capacity ceilings, or process ratios. It also does not track R&D spending, or financial/economic targets.	 • Since 2021, establish RMB 200 billion special refinancing loan to support the clean and efficient use of coal. • By 2025, increase the digitalization rate of production equipment to 55%.
5. Non-measurable / Ambiguous Targets	Vague target statements lacking clear definitions or involving unmeasurable metrics.	 • By 2025, installed capacity of new types of energy storage will reach 30 GW or more. • By 2025, renovation area of existing buildings will increase by >20 million m² compared to 2023. • By 2030, all ground vehicles and equipment at civil airports will strive to be powered by electricity.

After this filtering, we obtained 92 quantifiable national-level targets (with 2035 NDC and forest stock targets included) that could in principle be represented in TIAM. We then manually collected additional information for each target, such as governance level and policy coverage, and removed double counting at this stage, which yielded the final set of 47 targets included in the credibility assessment.

Regarding the impacts of exclusions, for reasons 1–4, we acknowledge that this TIAM model-based study may not fully capture the mitigation effects of these excluded targets, and we have highlighted this limitation in both the Discussion and SI section 1

Manuscript line 265-268:

“However, due to the quantification limits of TIAM-UCL, targets related to water, waste, recycling, air pollution, and climate finance (See SI table 2) are excluded from our modelling practice. As a result, our findings may underestimate the mitigation potential of these measures in GW and RS. Further quantitative research is needed to assess the decarbonisation potential of these targets.”

For reason 5, these targets are unlikely to be systematically implemented or tracked in practice. Their exclusion therefore does not alter results but reflect a systemic feature of Chinese climate policymaking: the prevalence of targets that lack measurability or enforceability. This provides additional qualitative insight for this paper. We’ve explicitly discussed this point in the Discussion.

Manuscript line 249-264:

“The policy gap is particularly evident in the absence of explicit, enforceable sectorial targets for fossil fuel phaseout. Vague policy signals such as “strictly limit” coal consumption or “strictly control” new coal projects weaken target credibility and allow discretionary interpretation at the local level³. Meanwhile, the prioritisation of domestic oil and gas production for energy security sends mixed signals, potentially reinforcing fossil fuel lock-in and diluting the consistency of China’s decarbonisation agenda (See SI Table 3). Further work is needed to examine how a more serious approach to coal phaseout could ensure system reliability, especially by scaling up viable alternatives for maintaining power system balance as coal declines.

Our policy review reveals that the weak enforcement partly stems from ambiguity in central policy design and deeper structural tensions in China’s climate policymaking and governance. For instance, industrial efficiency regulations define both basic and advanced performance standards but apply only to enterprises above a designated size, narrowing regulatory scope⁴. Even within the regulated group, the required share of firms meeting advanced standards rarely exceeds 50%, reflecting accommodation of regional development disparities (See SI_PolicyCredibilityAssessment.xlsx). Many enterprises continue to struggle to meet even the basic thresholds. This lack of clarity enables local governments to selectively interpret and implement policies based on local economic priorities. However, due to the quantification limits of TIAM-UCL, targets related to water, waste, recycling, air pollution, and climate finance (See SI table 2) are excluded from our modelling practice. As a result, our findings may underestimate the mitigation potential of these measures in GW and RS. Further quantitative research is needed to assess the decarbonisation potential of these targets.”

We hope this supplemented clarification could strengthen transparency and reinforce that the exclusion of certain targets does not undermine the robustness of our scenario results.

Comment 2.2

A second issue relates to the credibility assessment methodology. The framework and the choice of indicators (governance level, inclusion in the Five-Year Plan, and progress to date) are logical. There is a missed opportunity, however, to engage with literature discussing factors in the target implementation gap (e.g. 1, 2), which could provide a stronger foundation for the indicators chosen. Additionally, while the rationale for weighting progress to date higher than the other two indicators is clear, the weightings in general seem somewhat arbitrary. How sensitive are the results to the weightings chosen?

Response:

Thank you for this valuable comment. In the revised manuscript, we have engaged with the literature highlighted by the reviewer to provide a stronger basis for our choice of indicators

Manuscript lines 295-303:

“The definition of ‘policy credibility’ varies in the literature^{5,6}. Furthermore, there is no standard assessment framework for policy credibility across studies^{7,8}. Fransen et al. rely on cross-country composite indices such as the Climate Policy Score and the Climate Change Performance Index to capture policy adoption and outcome gaps⁹. Peterson and van Asselt propose five indicators to assess country-level NDC implementation risks: NDC ambition, institutional capacity, interest group opposition, policy inconsistency, and monitoring and enforcement, highlighting common challenges in closing the implementation gap¹⁰. While these frameworks are valuable for cross-country comparison, they are less suited to China’s context, where sectoral targets embedded in central plans are the primary drivers of emissions outcomes.”

We agree that weightings could appear arbitrary, and we therefore test the robustness of our results across five alternative weighting schemes (See SI Section 6.2):

- $\omega_{gov} : \omega_{FYP} : \omega_{TNPI} = 3 : 2 : 5$
- $\omega_{gov} : \omega_{FYP} : \omega_{TNPI} = 2 : 3 : 5$
- $\omega_{gov} : \omega_{FYP} : \omega_{TNPI} = 3 : 3 : 4$
- $\omega_{gov} : \omega_{FYP} : \omega_{TNPI} = 1 : 1 : 1$
- $\omega_{gov} : \omega_{FYP} : \omega_{TNPI} = 4 : 4 : 2$

Results show that our clusters are largely robust across five different weighting combinations. When the weight of the progress indicator is decreased relatively lower to the two governance indicators, two targets show rating uncertainty:

- *By 2025, renewable energy will account for about 18% of primary energy consumption (previously rated low in main case, now becomes medium).*
- *By 2030, electricity will account for 65% of building energy consumption (previously rated medium in main case, now becomes low).*

As a result, we generate a sensitivity case of **Calm Sea** scenario, denoted **CS_Low_TNPI**, which includes the first target but excludes the second. Since **Green Lights** includes all scenario clusters, it is unaffected in modelling practice.

Taken together (comments 2.1 and 2.2), we present China’s domestic sensitivity cases (GHG reduction targets for 2035 and the reduced TNPI weight case) in SI section 2.

SI Table 12. Sensitivity of credibility assessment to weighting schemes

Numerical Target	Main case	W_gov:W_FYP:W_Prog=3:2:5	_FYP:W_Prog=2:3:5	W_gov:W_FYP:W_Prog=3:3:4	W_gov:W_FYP:W_Prog=1:1:1	W_gov:W_FYP:W_Prog=4:4:2
By 2025, energy consumption per unit of GDP will be lowered by 13.5% from the 2020 level.	MEDIUM	MEDIUM	MEDIUM	MEDIUM	MEDIUM	MEDIUM
By 2025, carbon dioxide (CO ₂) emissions per unit of GDP will be lowered by 18% from the 2020 level.	MEDIUM	MEDIUM	MEDIUM	MEDIUM	MEDIUM	MEDIUM
By 2025, the share of non-fossil energy consumption will reach around 20%.	HIGH	HIGH	HIGH	HIGH	HIGH	HIGH
By 2025, the forest coverage rate will reach 24.1%, and the forest stock volume will rise to 18 billion cubic meters.	ACHIEVED	ACHIEVED	ACHIEVED	ACHIEVED	ACHIEVED	ACHIEVED
By 2025, electricity will account for about 30% of end-use energy consumption.	MEDIUM	MEDIUM	MEDIUM	MEDIUM	MEDIUM	MEDIUM
By 2025, total renewable energy consumption will reach about 1 billion tons of standard coal (~29,300 PJ).	MEDIUM	MEDIUM	MEDIUM	MEDIUM	MEDIUM	MEDIUM
By 2025, renewable energy will account for about 18% of primary energy consumption.	LOW	LOW	LOW	LOW	LOW	MEDIUM
By 2025, renewable energy will account for more than 50% of the incremental increase in primary energy consumption.	MEDIUM	MEDIUM	MEDIUM	MEDIUM	MEDIUM	MEDIUM
By 2025, the non-electric utilization of renewable energy (including geothermal heating, biomass heating, biomass fuels, and solar thermal utilization) will reach over 60 million tons of standard coal (~1,758 PJ).	MEDIUM	MEDIUM	MEDIUM	MEDIUM	MEDIUM	MEDIUM
By 2025, annual crude oil production will stabilize at around 200 million tons.	ACHIEVED	ACHIEVED	ACHIEVED	ACHIEVED	ACHIEVED	ACHIEVED
By 2025, annual natural gas production will exceed 230 billion cubic meters.	ACHIEVED	ACHIEVED	ACHIEVED	ACHIEVED	ACHIEVED	ACHIEVED
By 2025, 100,000–200,000 tons of hydrogen will be produced annually from renewable energy.	ACHIEVED	ACHIEVED	ACHIEVED	ACHIEVED	ACHIEVED	ACHIEVED
By 2025, total installed power generation capacity will reach approximately 3,000 GW.	ACHIEVED	ACHIEVED	ACHIEVED	ACHIEVED	ACHIEVED	ACHIEVED
By 2025, the proportion of non-fossil energy in power generation will reach about 39%.	MEDIUM	MEDIUM	MEDIUM	MEDIUM	MEDIUM	MEDIUM
By 2025, the operational installed capacity of nuclear power will reach approximately 70 GW.	ACHIEVED	ACHIEVED	ACHIEVED	ACHIEVED	ACHIEVED	ACHIEVED
By 2025, power generation from renewable energy will reach approximately 3,300 TWh.	ACHIEVED	ACHIEVED	ACHIEVED	ACHIEVED	ACHIEVED	ACHIEVED
By 2025, 33% of electricity will be generated from renewables.	ACHIEVED	ACHIEVED	ACHIEVED	ACHIEVED	ACHIEVED	ACHIEVED
By 2025, 18% of electricity will come from non-hydropower renewables.	ACHIEVED	ACHIEVED	ACHIEVED	ACHIEVED	ACHIEVED	ACHIEVED
By 2025, approximately 40 GW of additional hydropower capacity will be installed compared to 2020 (370 GW).	ACHIEVED	ACHIEVED	ACHIEVED	ACHIEVED	ACHIEVED	ACHIEVED
By 2025, CO ₂ emissions per unit of industrial added value will be reduced by 18%.	LOW	LOW	LOW	LOW	LOW	LOW
By 2025, energy consumption per unit of industrial added value in large-scale industries will decrease by 13.5% compared to 2020.	LOW	LOW	LOW	LOW	LOW	LOW
By 2025, the proportion of EAF in iron and steel production will increase to 15%.	LOW	LOW	LOW	LOW	LOW	LOW
By 2025, recycled steel use will reach 320 million tons (updated to 300 million tons in the latest 2024 policy).	LOW	LOW	LOW	LOW	LOW	LOW
By 2025, domestic primary crude oil refining capacity will be kept below 1 billion metric tons (20 million b/d), and the utilization rate of production capacity for main products will rise to 80% or more.	ACHIEVED	ACHIEVED	ACHIEVED	ACHIEVED	ACHIEVED	ACHIEVED
By 2025, comprehensive energy consumption per ton of steel will be reduced by over 2% compared to 2020 (updated in 2024 to a 2% reduction compared to 2023).	LOW	LOW	LOW	LOW	LOW	LOW
By 2025, electricity will account for approximately 30% of industrial end-use energy consumption.	LOW	LOW	LOW	LOW	LOW	LOW
By 2025, the carbon intensity of road transportation will be reduced by 5% compared to 2020.	ACHIEVED	ACHIEVED	ACHIEVED	ACHIEVED	ACHIEVED	ACHIEVED
By 2025, the average electricity consumption of new passenger BEVs will be ≤12.0 kWh/100 km.	ACHIEVED	ACHIEVED	ACHIEVED	ACHIEVED	ACHIEVED	ACHIEVED
By 2025, the average fuel consumption of new passenger cars will be reduced to 4.0 liters/100 km.	ACHIEVED	ACHIEVED	ACHIEVED	ACHIEVED	ACHIEVED	ACHIEVED
By 2025, new energy buses will account for 72% of all surface public transport vehicles in urban areas.	ACHIEVED	ACHIEVED	ACHIEVED	ACHIEVED	ACHIEVED	ACHIEVED
By 2025, sales of new energy vehicles will account for about 20% of total new car sales.	ACHIEVED	ACHIEVED	ACHIEVED	ACHIEVED	ACHIEVED	ACHIEVED
By 2025, electricity consumption will account for over 55% of building energy consumption.	ACHIEVED	ACHIEVED	ACHIEVED	ACHIEVED	ACHIEVED	ACHIEVED
By 2025, renewable energy use in urban buildings will reach 8%.	MEDIUM	MEDIUM	MEDIUM	MEDIUM	MEDIUM	MEDIUM
By 2030, CO ₂ emissions per unit of GDP will be reduced by over 65% from the 2005 level.	HIGH	HIGH	HIGH	HIGH	HIGH	HIGH
By 2030, the share of non-fossil fuels in primary energy consumption will reach around 25%.	HIGH	HIGH	HIGH	HIGH	HIGH	HIGH
By 2030, the forest coverage rate will reach about 25%, and the forest stock volume will reach 19 billion cubic meters.	ACHIEVED	ACHIEVED	ACHIEVED	ACHIEVED	ACHIEVED	ACHIEVED
By 2030, approximately 40 GW of additional hydropower capacity will be installed compared to 2025 (370 GW).	HIGH	HIGH	HIGH	HIGH	HIGH	HIGH
By 2030, the total installed capacity of wind and solar power will exceed 1,200 GW.	ACHIEVED	ACHIEVED	ACHIEVED	ACHIEVED	ACHIEVED	ACHIEVED
By 2030, around 40% of incremental vehicles will be fueled by new and clean energy.	HIGH	HIGH	HIGH	HIGH	HIGH	HIGH
By 2030, petroleum consumption for land transportation will peak before 2030.	HIGH	HIGH	HIGH	HIGH	HIGH	HIGH
By 2030, the average fuel consumption of new passenger cars will be reduced to 3.2 liters/100 km.	ACHIEVED	ACHIEVED	ACHIEVED	ACHIEVED	ACHIEVED	ACHIEVED
By 2030, 100% of public vehicles will be electrified.	HIGH	HIGH	HIGH	HIGH	HIGH	HIGH
By 2030, the proportion of EAF in iron and steel production will increase to over 20%.	LOW	LOW	LOW	LOW	LOW	LOW
By 2030, electricity will account for over 65% of building energy consumption.	MEDIUM	MEDIUM	MEDIUM	MEDIUM	MEDIUM	LOW
By 2035, the share of non-fossil fuels in total energy consumption will exceed 30%.	HIGH	HIGH	HIGH	HIGH	HIGH	HIGH
By 2035, the total installed capacity of wind and solar power generation will reach 3,600 GW, more than six times the 2020 level.	HIGH	HIGH	HIGH	HIGH	HIGH	HIGH
By 2060, the share of non-fossil fuels in energy consumption will exceed 80%.	LOW	LOW	LOW	LOW	LOW	LOW

Comment 2.3

A third issue concerns the treatment of targets after their stated target year. While the paper is clear about which targets are assumed to be achieved in which scenarios, it is less clear how this achievement is translated into the modelling framework, and in particular what was assumed about progress on the target indicators following the target year. There are two potentially conflicting logics: On the one hand, one could interpret an “achieved” target as evidence of strong institutional capacity and assume continued progress thereafter. Alternatively, an “achieved” target could reflect an unambitious goal, which might correlate with weak implementation capacity, in which case it would not signal a trajectory of sustained improvement. I was not able to decipher how such logic is carried forward from the credibility assessment into the scenario narratives and ultimately into the IAM results. Greater clarity on this point would be valuable.

Response:

Thank you for this insightful comment. In our framework, policy clusters are implemented stepwise from Great Wall to Green Lights (i.e., achieved, achieved + high, achieved + high + medium, etc.). For targets that are already achieved or even overachieved, we implement the actual realised levels by 2025 rather than the stated target levels to better reflect China’s existing climate efforts. For example, this applies to indicators such as EV market share or installed solar and wind capacity.

In modelling practice, we assume that these achieved levels will not decline after the target year; instead, they are maintained as the minimum baseline from 2025 onwards. As additional targets are implemented from Great Wall to Green Lights, some previously achieved targets are further raised to align with higher-level ambition, normative-based goals (e.g., non-fossil share, net-zero commitment). A clear example can be seen in the Calm Sea and Green Lights scenarios: once the net-zero target is introduced, the “achieved” policy levers define the lower bound of the energy system structure (e.g., sectoral electrification level), while the net-zero and other more system-wide targets further strengthen these levers to ensure that the overall pathway is consistent with achieving net zero. This also helps identify the policy gap between the policies currently issued and the commitments made under the net-zero pledge. We have added more clarity on this point in the SI line 292-299.

Comment 2.4

A fourth set of issues relates to the global net zero scenario. The accuracy of assumptions in Table 2 should be confirmed. For instance, while emission coverage for Central and South America is said to cover CO₂ only, the net-zero targets of the largest countries in the region (e.g. Brazil, Argentina, Chile, Peru, Colombia) cover all GHGs (3, Likewise for the former Soviet Union. The basis for choosing a “representative” country for certain regions should

also be clarified – are these countries representative in terms of size of economy, emissions profile, or some other indicator? (For example, on what basis is Nigeria said to be representative of Africa?) Alternatively, it is plausible that the results are not particularly sensitive to these assumptions, in which case, it would be helpful to clarify that.

References

1. L. Peterson, H. and van Asselt, Assessing risks to the implementation of NDCs under the Paris Agreement. *Clim. Policy* 0, 1–15.
2. T. Fransen, J. Meckling, A. Stünzi, T. S. Schmidt, F. Egli, N. Schmid, C. Beaton, Taking stock of the implementation gap in climate policy. *Nat. Clim. Change*, 1–4 (2023).
3. Climate Watch, Net-Zero Tracker (2024). <https://www.climatewatchdata.org/net-zero-tracker>.
4. Energy and Climate Intelligence Unit, Data Driven Envirolab, New Climate Institute, University of Oxford, Net Zero Tracker, (2024); <https://zerotracker.net/>.

Response:

We thank the reviewer for highlighting this issue related to our earlier regional net-zero target assumptions and for directing us toward country-level emissions and Net Zero Tracker data.

For TIAM regions with diverse national pledges (AFR, CSA, FSU, MEA, ODA), we revisited all relevant countries and compiled their 2020 historical GHG emissions and regional shares from Climate Watch (<https://www.climatewatchdata.org>), along with stated net-zero targets and coverage (CO₂-only vs. all GHGs) from the Net Zero Tracker (<https://zerotracker.net>), with data updated to 1 August 2025 (See Response Table 1). We then filtered for countries with net-zero targets and aggregated their pledged emissions for 2050, 2060, and 2070 to assess the proportion of each region’s 2020 emissions committed to net zero (see Manuscript Table 3.b). Instead of imposing uniform regional net-zero constraints, we assume that each region partially achieves net zero based on the share of countries within it that have adopted net-zero targets and converge to net zero GHG emissions by 2100. The share of emissions contributed by countries within each region is held constant from 2020 onward. This approach provides a more realistic representation of current ambition levels and likely mitigation outcomes in developing regions.

The new regional net-zero target construction approach has been updated in manuscript, with full data sources and calculations documented in *TIAM_region_net_zero_target_assumptions.xlsx*.

Table 1. Net-zero pledges by country across TIAM-selected regions

TIAM region	Country	2020 GHG emissions (MtCO ₂ e, approx)	Share of region total	End target	NetZero/Target Year	GHG Coverage

AFR	Democratic Republic of the Congo	707	16.65%	Emissions reduction target	2030	Not Specified
AFR	South Africa	525.06	12.36%	Net zero	2050	Carbon dioxide and other GHGs
AFR	Nigeria	356.46	8.39%	Net zero	2060	Carbon dioxide and other GHGs
AFR	Egypt	325.48	7.66%	Other	2030	Carbon dioxide and other GHGs
AFR	Ethiopia	219.58	5.17%	Net zero	2050	Carbon dioxide and other GHGs
AFR	Tanzania	174.96	4.12%	Net zero	2050	Carbon dioxide and other GHGs
AFR	Angola	139.08	3.28%	Net zero	2050	Carbon dioxide and other GHGs
AFR	Sudan	133.48	3.14%	Net zero	2050	Carbon dioxide and other GHGs
AFR	Chad	114.31	2.69%	Emissions reduction target	2030	Carbon dioxide and other GHGs
AFR	Mozambique	112.05	2.64%	Net zero	2050	Not Specified
AFR	Zimbabwe	111.37	2.62%	Emissions reduction target	2030	Carbon dioxide and other GHGs
AFR	Zambia	99.06	2.33%	Reduction v. BAU	2030	Carbon dioxide and other GHGs
AFR	Morocco	95.5	2.25%	Reduction v. BAU	2030	Carbon dioxide and other GHGs
AFR	Kenya	84.4	1.99%	Reduction v. BAU	2035	Carbon dioxide and other GHGs
AFR	Cameroon	76.55	1.80%	Emissions reduction target	2030	Carbon dioxide and other GHGs
AFR	South Sudan	74.45	1.75%	Emissions reduction target	2030	Carbon dioxide and other GHGs
AFR	Libya	68.82	1.62%	No target	0	Not Specified
AFR	Côte d'Ivoire	67.55	1.59%	Reduction v. BAU	2030	Carbon dioxide and other GHGs
AFR	Uganda	62	1.46%	Net zero	2030	Carbon dioxide and other GHGs
AFR	Burkina Faso	54.48	1.28%	Net zero	2050	Carbon dioxide and other GHGs

AFR	Botswana	53.94	1.27%	Emissions reduction target	2030	Carbon dioxide and other GHGs
AFR	Central African Republic	53.52	1.26%	Net zero	2050	Carbon dioxide and other GHGs
AFR	Mali	46.95	1.11%	Net zero	2050	Carbon dioxide and other GHGs
AFR	Somalia	46.64	1.10%	Reduction v. BAU	2035	Carbon dioxide and other GHGs
AFR	Niger	42.64	1.00%	Net zero	2050	Not Specified
AFR	Madagascar	42.13	0.99%	Net zero	2050	Carbon dioxide and other GHGs
AFR	Guinea	40.18	0.95%	Net zero	2050	Carbon dioxide and other GHGs
AFR	Tunisia	39.2	0.92%	Carbon neutral(ity)	2050	Carbon dioxide only
AFR	Senegal	32.8	0.77%	Emissions reduction target	2030	Carbon dioxide and other GHGs
AFR	Congo	30.69	0.72%	Emissions reduction target	2030	Carbon dioxide and other GHGs
AFR	Benin	27.37	0.64%	Reduction v. BAU	2030	Carbon dioxide and other GHGs
AFR	Namibia	23.46	0.55%	Net zero	2050	Carbon dioxide and other GHGs
AFR	Gabon	22.11	0.52%	Carbon neutral(ity)	2050	Carbon dioxide and other GHGs
AFR	Malawi	19.95	0.47%	Net zero	2050	Carbon dioxide and other GHGs
AFR	Mauritania	17.99	0.42%	Carbon neutral(ity)	2030	Carbon dioxide and other GHGs
AFR	Ghana	16.93	0.40%	Net zero	2060	Carbon dioxide and other GHGs
AFR	Liberia	14.95	0.35%	Net zero	2050	Carbon dioxide and other GHGs
AFR	Equatorial Guinea	13.11	0.31%	Emissions reduction target	2050	Carbon dioxide and other GHGs
AFR	Sierra Leone	9.57	0.23%	Emissions reduction target	2050	Carbon dioxide and other GHGs
AFR	Togo	9.34	0.22%	Net zero	2050	Carbon dioxide and other GHGs

AFR	Rwanda	9.32	0.22%	Carbon neutral(ity)	2050	Carbon dioxide and other GHGs
AFR	Eritrea	6.88	0.16%	Reduction v. BAU	2030	Carbon dioxide and other GHGs
AFR	Burundi	6.57	0.15%	Emissions reduction target	2050	Carbon dioxide and other GHGs
AFR	Guinea-Bissau	4.56	0.11%	Emissions reduction target	2030	Carbon dioxide and other GHGs
AFR	Gambia	3.92	0.09%	Net zero	2050	Carbon dioxide and other GHGs
AFR	Lesotho	3.21	0.08%	Net zero	2050	Carbon dioxide and other GHGs
AFR	Eswatini	2.87	0.07%	Reduction v. BAU	2030	Carbon dioxide and other GHGs
AFR	Djibouti	1.53	0.04%	Net zero	2050	Carbon dioxide only
AFR	Seychelles	0.86	0.02%	Net zero	2050	Carbon dioxide and other GHGs
AFR	Cabo Verde	0.6	0.01%	Net zero	2050	Carbon dioxide and other GHGs
AFR	Comoros	0.58	0.01%	Net zero	2050	Carbon dioxide and other GHGs
AFR	São Tomé and Príncipe	0.4	0.01%	Net zero	2050	Carbon dioxide and other GHGs
CSA	Brazil	1537.1	45.64%	Carbon neutral(ity)	2050	Carbon dioxide and other GHGs
CSA	Argentina	394.51	11.71%	Net zero	2050	Carbon dioxide and other GHGs
CSA	Colombia	292.4	8.68%	Carbon neutral(ity)	2050	Carbon dioxide and other GHGs
CSA	Venezuela	240.16	7.13%	Emissions reduction target	2030	Carbon dioxide and other GHGs
CSA	Peru	184.48	5.48%	Net zero	2050	Carbon dioxide and other GHGs
CSA	Bolivia	133.47	3.96%	No target	0	Not Specified
CSA	Paraguay	92.82	2.76%	Reduction v. BAU	2030	Carbon dioxide and other GHGs
CSA	Ecuador	91.42	2.71%	Emissions reduction target	2035	Not Specified

CSA	Chile	50.08	1.49%	Carbon neutral(ity)	2050	Carbon dioxide and other GHGs
CSA	Guatemala	42.57	1.26%	Emissions reduction target	2030	Carbon dioxide and other GHGs
CSA	Nicaragua	39.49	1.17%	Net zero	2050	Carbon dioxide and other GHGs
CSA	Dominican Republic	39.26	1.17%	Net zero	2050	Carbon dioxide and other GHGs
CSA	Uruguay	33.29	0.99%	Net zero	2050	Carbon dioxide and other GHGs
CSA	Cuba	28.55	0.85%	Other	2050	Carbon dioxide and other GHGs
CSA	Honduras	27.49	0.82%	Reduction v. BAU	2030	Carbon dioxide and other GHGs
CSA	Trinidad and Tobago	25.29	0.75%	Emissions reduction target	2030	Carbon dioxide and other GHGs
CSA	Panama	23.62	0.70%	Net zero	2050	Carbon dioxide and other GHGs
CSA	Guyana	18.83	0.56%	Net zero	2050	Not Specified
CSA	Suriname	13.1	0.39%	Net zero	2050	Carbon dioxide and other GHGs
CSA	El Salvador	12.85	0.38%	Absolute emissions target	2030	Carbon dioxide and other GHGs
CSA	Haiti	11.61	0.34%	Net zero	2050	Not Specified
CSA	Jamaica	10.35	0.31%	Net zero	2050	Not Specified
CSA	Costa Rica	7.42	0.22%	Net zero	2050	Carbon dioxide and other GHGs
CSA	Belize	6.69	0.20%	Net zero	2050	Carbon dioxide and other GHGs
CSA	Barbados	3.83	0.11%	Net zero	2035	Carbon dioxide and other GHGs
CSA	Antigua and Barbuda	3.58	0.11%	Net zero	2040	Carbon dioxide and other GHGs
CSA	Bahamas	2.25	0.07%	Emissions reduction target	2030	Carbon dioxide and other GHGs
CSA	Saint Lucia	0.42	0.01%	Reduction v. BAU	2030	Carbon dioxide and other GHGs
CSA	Grenada	0.35	0.01%	Emissions reduction target	2030	Carbon dioxide and other GHGs

CSA	Saint Vincent and the Grenadines	0.28	0.01%	Net zero	2050	Carbon dioxide and other GHGs
CSA	Saint Kitts and Nevis	0.26	0.01%	Net zero	2050	Carbon dioxide only
CSA	Dominica	0.22	0.01%	Carbon neutral(ity)	2030	Carbon dioxide only
FSU	Russia	1759.19	59.18%	Carbon neutral(ity)	2060	Carbon dioxide and other GHGs
FSU	Kazakhstan	318.77	10.72%	Carbon neutral(ity)	2060	Carbon dioxide and other GHGs
FSU	Ukraine	246.31	8.29%	Carbon neutral(ity)	2060	Carbon dioxide and other GHGs
FSU	Turkmenistan	204.84	6.89%	Emissions reduction target	2030	Carbon dioxide and other GHGs
FSU	Uzbekistan	177.87	5.98%	Carbon neutral(ity)	2050	Carbon dioxide and other GHGs
FSU	Belarus	91.67	3.08%	Emissions reduction target	2030	Carbon dioxide and other GHGs
FSU	Azerbaijan	50.07	1.68%	Emissions reduction target	2050	Carbon dioxide and other GHGs
FSU	Lithuania	25.02	0.84%	Climate neutral	2050	Carbon dioxide and other GHGs
FSU	Tajikistan	17.59	0.59%	Emissions reduction target	2030	Carbon dioxide and other GHGs
FSU	Georgia	17.07	0.57%	GHG neutral(ity)	2050	Carbon dioxide and other GHGs
FSU	Estonia	15.06	0.51%	Net zero	2050	Carbon dioxide and other GHGs
FSU	Kyrgyzstan	14.06	0.47%	Carbon neutral(ity)	2050	Not Specified
FSU	Moldova	13.38	0.45%	Carbon neutral(ity)	2050	Carbon dioxide and other GHGs
FSU	Latvia	11.34	0.38%	Carbon neutral(ity)	2050	Carbon dioxide and other GHGs
FSU	Armenia	10.6	0.36%	Climate neutral	2050	Carbon dioxide and other GHGs
MEA	Iran	909.92	30.19%	Emissions reduction target	2030	Carbon dioxide and other GHGs

MEA	Saudi Arabia	725.04	24.05%	Net zero	2060	Carbon dioxide and other GHGs
MEA	Turkey	474.98	15.76%	Net zero	2053	Carbon dioxide and other GHGs
MEA	United Arab Emirates	242.06	8.03%	Net zero	2050	Carbon dioxide and other GHGs
MEA	Qatar	121.82	4.04%	Emissions reduction target	2030	Carbon dioxide and other GHGs
MEA	Kuwait	121.39	4.03%	Carbon neutral(ity)	2060	Not Specified
MEA	Oman	95.24	3.16%	Net zero	2050	Carbon dioxide and other GHGs
MEA	Israel	81.81	2.71%	Net zero	2050	Not Specified
MEA	Syria	77.14	2.56%	No target	0	Not Specified
MEA	Bahrain	57.33	1.90%	Net zero	2060	Not Specified
MEA	Yemen	30.08	1.00%	Net zero	2030	Carbon dioxide and other GHGs
MEA	Jordan	29.68	0.98%	Reduction v. BAU	2030	Carbon dioxide and other GHGs
MEA	Lebanon	26.99	0.90%	Net zero	2050	Not Specified
MEA	Brunei	12.63	0.42%	Net zero	2050	Carbon dioxide and other GHGs
MEA	Cyprus	8.05	0.27%	Climate neutral	2050	Carbon dioxide and other GHGs
ODA	Indonesia	1449.43	31.57%	Net zero	2060	Carbon dioxide and other GHGs
ODA	Pakistan	578.89	12.61%	Net zero	2050	Carbon dioxide and other GHGs
ODA	Vietnam	473.76	10.32%	Net zero	2050	Carbon dioxide and other GHGs
ODA	Thailand	414.29	9.02%	Net zero	2065	Carbon dioxide and other GHGs
ODA	Malaysia	371.93	8.10%	Net zero	2050	Carbon dioxide and other GHGs
ODA	Bangladesh	242.74	5.29%	Net zero	2030	Carbon dioxide and other GHGs
ODA	Philippines	235.96	5.14%	Reduction v. BAU	2030	Carbon dioxide and other GHGs
ODA	Myanmar	231.83	5.05%	Net zero	2050	Carbon dioxide only

ODA	North Korea	88.66	1.93%	Reduction v. BAU	2030	Carbon dioxide and other GHGs
ODA	Cambodia	80.35	1.75%	Net zero	2050	Carbon dioxide and other GHGs
ODA	Mongolia	76.63	1.67%	Emissions reduction target	2030	Carbon dioxide and other GHGs
ODA	Singapore	59.8	1.30%	Net zero	2050	Carbon dioxide and other GHGs
ODA	Nepal	58.45	1.27%	Net zero	2045	Carbon dioxide only
ODA	Papua New Guinea	50.15	1.09%	Carbon neutral(ity)	2050	Carbon dioxide only
ODA	Solomon Islands	45.82	1.00%	Net zero	2050	Carbon dioxide and other GHGs
ODA	Laos	45.35	0.99%	Net zero	2050	Carbon dioxide and other GHGs
ODA	Sri Lanka	39.53	0.86%	Carbon neutral(ity)	2050	Carbon dioxide and other GHGs
ODA	Afghanistan	32.08	0.70%	Reduction v. BAU	2030	Carbon dioxide and other GHGs
ODA	Mauritius	6.43	0.14%	Carbon neutral(ity)	2070	Not Specified
ODA	East Timor	5.57	0.12%	Net zero	2050	Carbon dioxide and other GHGs
ODA	Maldives	2.13	0.05%	Absolute emissions target	2035	Carbon dioxide and other GHGs
ODA	Bhutan	0.7	0.02%	Carbon negative	2050	Carbon dioxide and other GHGs
ODA	Vanuatu	0.53	0.01%	Net zero	2050	Carbon dioxide and other GHGs
ODA	Samoa	0.5	0.01%	Net zero	2050	Carbon dioxide and other GHGs
ODA	Tonga	0.3	0.01%	Net zero	2050	Carbon dioxide and other GHGs
ODA	Kiribati	0.09	0.00%	Net zero	2050	Carbon dioxide only
ODA	Fiji	-0.86	-0.02%	Net zero	2050	Carbon dioxide and other GHGs

Manuscript Table 3. (b) Net-zero pledge coverage rate and assumptions for TIAM selected region

	2050	2060	2070	2100
Africa	42.71%	52.10%	52.1%	100%
Central and South America	72.29%	72.29%	72.29%	100%
Former Soviet Union	8.37%	86.55%	86.55%	100%
Middle East	31.98%	61.96%	61.96%	100%
Other Developing Asia	49.78%	81.35%	90.5%	100%

Reference

1. He, Y., Piao, S., Ciais, P., Xu, H. & Gasser, T. Future land carbon removals in China consistent with national inventory. *Nat. Commun.* **15**, 10426 (2024).
2. Pye, S. *et al.* The TIAM-UCL Model (Version 4.1.1) Documentation. (2020).
3. CCP & State Council. China's Achievements, New Goals and New Measures for Nationally Determined Contributions. (2021).
4. NDRC. Action Plan for Strict Energy Efficiency Constraints to Promote Energy Conservation and Carbon Reduction in Key Industries (2021-2025).
https://www.ndrc.gov.cn/xxgk/zcfb/tz/202110/t20211021_1300583_ext.html (2021).
5. Brunner, S., Flachslund, C. & Marschinski, R. Credible commitment in carbon policy. *Clim. Policy* **12**, 255–271 (2012).
6. Jacobs, A. M. Policy Making for the Long Term in Advanced Democracies. *Annu. Rev. Polit. Sci.* **19**, 433–454 (2016).
7. Victor, D. G., Lumkowsky, M. & Dannenberg, A. Determining the credibility of commitments in international climate policy. *Nat. Clim. Change* **12**, 793–800 (2022).
8. Teng, F. Ambitious and credible pledges. *Nat. Clim. Change* **12**, 779–780 (2022).
9. Fransen, T. *et al.* Taking stock of the implementation gap in climate policy. *Nat. Clim. Change* **13**, 752–755 (2023).

10. Peterson, L. & van Asselt, H. Assessing risks to the implementation of NDCs under the Paris Agreement. *Clim. Policy* **0**, 1–15.

Responses to Reviewers

NCOMMS-25-41170A

December 2025

REVIEWER COMMENTS

Reviewer #1 (Remarks to the Author):

In this revised version, the authors have substantially strengthened the manuscript and have addressed most of the concerns raised in the previous review. The paper now presents a timely and policy-relevant assessment of China's climate policy credibility and its global implications, combining a structured policy review with scenario-based integrated assessment modelling. I have only a few remaining minor points—mainly clarifications for transparency and small figure/presentation refinements—that would further improve readability and reproducibility before publication.

Response:

Thank you for your positive and constructive assessment of the revised manuscript. We are pleased that the revisions have addressed most of the concerns raised and strengthened the policy relevance and clarity of the paper. We have carefully addressed the remaining points to further improve transparency, readability, and reproducibility. The corresponding revision and responses are presented below.

1. The manuscript/abstract could be read as implying that the structured credibility assessment was applied to all identified targets (e.g., 292 targets across policy documents). However, the Supplementary Information indicates a multi-step filtering process: 292 numerical targets were identified, then a subset of “quantifiable” targets was retained (reported as 90, including duplicates), which was further consolidated to a final set of 47 targets that were actually evaluated/scored for credibility. Please clarify this explicitly in the main text/abstract to avoid confusion—e.g., state that the study identifies 292 targets but the credibility scoring is conducted on the filtered set of quantifiable targets (final N=47), and briefly explain the filtering rationale.

Response:

Thank you for this suggestion. We have clarified the policy filtering and quantification process in both the Introduction and Methods to avoid any misunderstanding.

In the Introduction (Lines 40-42):

“A total of 292 numerical targets across 58 policy documents are reviewed. Of these, 47 quantifiable targets are filtered and implemented in the energy system model (see Methods).”

In the Methods (Lines 314-321):

“In total, 58 policy documents were analysed, covering 292 numerical targets, including China’s updated 2035 NDC.

Given the quantification limits of the TIAM-UCL model and the fact that some targets are unmeasurable (e.g. all ground vehicles and equipment at civil airports will strive to be powered by electricity by 2030), 47 targets were filtered and included in our credibility assessment (See Figure 3. (b)). The detailed reasons and implications for target non-quantifiability are discussed in SI Section 1. Target-level details are provided in the Supplementary Data. In the context of China, we developed and applied a credibility rating based on three key policy characteristics:”

2. While the manuscript states that scenario narratives are informed by the policy credibility assessment, the mapping between individual policy credibility scores and the quantified scenario assumptions remains insufficiently specified. This may lead to confusion for non-specialist readers regarding the purpose of the credibility assessment, how it influences scenario construction, and how it ultimately affects the modeling results. Given that relevant tables and detailed mappings already appear to be provided in the Supplementary Information, the authors are encouraged to add a brief explanatory bridge in the Introduction and/or Methods sections.

Response:

Thank you for your suggestions. To better clarify the policy quantification process and the key underlying assumptions, and in line with the manuscript’s conciseness requirements, we added an additional subsection in the Methods section entitled “Policy quantification across Chinese energy transition scenarios” and cross-refer to the relevant details in the SI.

Lines 476-488:

“Policy quantification is guided by the assessed credibility of individual targets—classified as Achieved, High, Medium, or Low; and by scenario narratives developed using a morphological approach. The Great Wall scenario, which enforces only Achieved targets, therefore represents the most conservative pathway, reflecting minimal policy delivery. For targets that have already been achieved or overachieved, we implement the realised levels observed by 2025 rather than the stated target values, in order to more accurately capture China’s existing climate efforts. Beyond the target year, these achieved levels are assumed to be maintained and form a minimum baseline from 2025 onwards.

Across scenarios, policy implementation expands progressively from higher- to lower-credibility targets. Moving from Great Wall to Green Lights, the assumed strength of policy delivery increases, with an expanding set of targets implemented, ultimately culminating in the full realisation of China’s overarching climate commitment: carbon neutrality by 2060 or earlier (assumed as 2050). Detailed mappings between specific targets and scenarios are provided in SI Table 6..”

3. The authors are encouraged to carefully check the consistency and completeness of cross-referencing between the main text and the Supplementary Information (SI). At present, several tables and figures in the SI appear not to be explicitly cited in either the main text or within the SI itself. For example, in the SI (around line 140), the text refers to “(see SI Table X, Step 3)”, the corresponding table reference is unclear. In addition, SI Tables 4, 5, 7, 8, 11, and 12 do not appear to be cited anywhere in the main manuscript or the SI text. A similar issue applies to several supplementary figures.

Response:

Thank you for your careful review, and we apologise for this oversight. We have now thoroughly checked the consistency and completeness of cross-referencing between the main text and the Supplementary Information. All SI tables and figures are now explicitly cited in the main manuscript, and unclear references (e.g., “see SI Table X, Step 3”) have been corrected.

In particular, SI Tables 4, 5, 7, 8, and 11–12 are now cross-referenced in the Figure 4 caption, which has been revised to read from lines 467-471:

“To quantify the scenario narratives in TIAM-UCL, we examine five key metrics and their associated uncertainties in SI Section 4, along with the corresponding modelling assumptions: national climate policy delivery (SI Section 4.1, Tables 5 and 6), GDP growth (SI Section 4.2, Figure 3), population(SI Section 4.3, Tables 7 and 8), clean technology cost reductions(SI Section 4.4, Table 9), and CCS deployment rates(SI Section 4.5, Table 10).”

4. Figure 1(b): The dashed line indicating the share of renewable energy (black) overlaps with the coal bar for the year 2020, which is also shown in black. This makes it difficult to identify the renewable share value for that year.

Response:

Thank you for pointing this out. We changed the colour of the dashed line in Figure 1(b) to avoid overlap with the coal bar in 2020. The revised figure has been updated in the manuscript.

5. Figures 2(a) and 2(b): In the GW scenario, the grey uncertainty band overlaps with the red shaded area representing the RS scenario, resulting in a bluish color that is visually ambiguous. Adjusting the plotting order (layering) or modifying transparency/color choices would help avoid color blending and improve interpretability.

Response:

Thank you for raising this point. We have revised Figures 2(a) and 2(b) by adjusting the transparency and plotting order, and by increasing the colour contrast to avoid colour blending between the GW uncertainty band and the RS shaded area. The revised figures are now clearer and easier to interpret.

6. Figure 3(a): Some text elements are obscured or truncated. On the left-hand side, labels are partially covered by lines, while on the right-hand side the word “recommendation” appears broken or cut off. Please revise the layout to ensure all text is fully visible and legible.

Response:

Thank you for your suggestion. We’ve adjusted the text and layout as much as possible to avoid text broken or cut off issue. New revised figure 3(a) now is fully visible.

7. Figure 3(b): (1) The governance level indicator (from the CCP to ministries) is currently encoded solely through bubble size. However, the visual contrast between sizes appears insufficient to clearly distinguish different administrative levels, particularly between the largest bubbles (CCP & State Council) and the medium-sized bubbles (e.g. NDRC). The authors may consider combining symbol shape with size to enhance visual differentiation—for example, using large circles for CCP-level targets and smaller symbols (e.g. triangles or stars) for targets issued by the NDRC and ministries.

(2) The figure contains a large number of data points and employs multiple discontinuous or non-linear axis segments, which results in substantial crowding in key regions (e.g., credibility scores between ~0.7 and 1.1). This leads to overlapping labels and makes it difficult to identify individual policy targets. The authors may wish to explore alternative visualization strategies, such as segmented or logarithmic scaling, selective zoom-in in critical ranges, or inset panels focusing on dense regions, to improve legibility.

(3) In light of the scenario narratives developed in this section, the figure could be further strengthened by explicitly highlighting or grouping key clusters of targets. For example, distinguishing targets with high credibility scores but lagging progress (which may require strong policy reinforcement as represented in the GL scenario), versus targets with low credibility yet high mitigation relevance (which may be deprioritized under the GW scenario), would enhance the narrative coherence between the credibility assessment and the subsequent scenario analysis.

Response:

Thank you for your constructive suggestions.

(1)–(2), we tested all of the approaches you proposed. However, some options increased visual complexity and reduced overall interpretability (e.g., logarithmic scaling and zoom-ins/inset views for dense ranges). We therefore implemented a combination of symbol

shape and bubble size to better differentiate governance levels. In addition, we refined the axis break from 0.3-1.2, increased the relative space allocated to the crowded mid-range, and slightly enlarged the overall figure to improve legibility.

(3) To strengthen the link between the credibility assessment and the scenario narratives, we added credibility thresholds to group targets into clusters and highlighted the achieved targets in grey. We also expanded the figure caption to clarify how the figure should be interpreted and how the identified clusters relate to the scenario narratives and quantification. The revised figure and caption have been incorporated into the manuscript.

Figure 3.b Credibility assessment of China’s climate and energy goals across sectors (details of each target assessment are provided in SI Section 2). Each shaped bubble represents a policy target. Bubble colour indicates the sector affected, and bubble shape reflects the level of government issuing the target (from the CCP and State Council to ministries, i.e., from higher to lower authority). The x-axis shows the raw progress towards each target (as of October 2025), and the y-axis reports the final credibility score, calculated as a weighted average of the three evaluation criteria described in the text ($\omega_{gov} : \omega_{FYP} : \omega_{TNPI} = 3 : 2 : 5$). The grey region indicates targets that had been achieved or overachieved by the review date and are therefore treated as implemented from the GW scenario onwards. The dashed lines mark the credibility thresholds: high credibility (score \geq the 67th percentile of all unachieved targets) and medium credibility (33rd–67th percentile).

8. Line 356: The text states “Where B_v,T_v,T_v are the base, target, and current years”, but this should likely write B_y,T_y,T_y.

Response:

Thank you for pointing this out, and we apologise for the oversight. We have corrected this notation and have also checked and revised the corresponding text in the Supplementary Information where relevant.

9. Line 365-366: While the manuscript states that the credibility score combines three components and tests five alternative weighting schemes (SI Section 1), the main text does not specify the actual weights applied to each indicator, nor clarify which weighting scheme(s) underpin the results shown in Fig. 3 and the subsequent scenario analysis. Given that the credibility score plays a central role in informing scenario narratives, a brief description of the weighting logic and reference to the relevant SI tables in the main text would improve transparency and reproducibility.

Response:

Thank you for this suggestion. We have clarified the weighting scheme used to construct the credibility score in the Figure 3 caption (as shown above) and both the main text (Lines 371-381)

“For targets still in progress (raw progress < 1), we classify them using a weighted credibility score. Each target’s score combines three components: governance level, inclusion in the 14th Five-Year Plan, and the TNPI. Figure 3. (b) presents the overall credibility score using weights where $\omega_{gov}:\omega_{FYP}:\omega_{TNPI}=3:2:5$. Targets are then clustered into categories as follows: High credible (score \geq 67th percentile), Medium credible (33rd–67th percentile), and Low credible (below 33rd percentile). To evaluate the sensitivity of credibility scores to weighting scheme and ensure the robustness of the clustering, we tested five alternative weighting schemes (see SI Section 1). Across weighting schemes, target classifications are largely robust. Only two targets shift category when the TNPI weight is lower than the other two indicators (see SI Table 13). We therefore adopt the consistent clustering results from the first four weighting schemes for our subsequent scenario quantification and report an additional weighting-related sensitivity case in SI Section 6.1. SI Figure 5 and 6 indicate that weighting-related sensitivity does not undermine our findings on China’s energy transition and its emission implications.”

10. The analysis focuses on 47 quantifiable targets due to model constraints and data availability. While this filtering is understandable, it may introduce a systematic selection bias by excluding non-quantifiable but policy-relevant measures (e.g., water, waste, recycling, air pollution control, and detailed industrial or building efficiency policies). As a result, mitigation contributions from easily measurable sectors such as power and transport may be overstated, which may shift sectoral emphasis and alter perceived feasibility/costs; please discuss likely direction and (if possible) sensitivity. The authors are encouraged to discuss how this sectoral imbalance may affect the estimated feasibility, costs, and

structural characteristics of China’s net-zero transition, and to clarify the likely direction of the bias introduced.

Response:

Thank you for raising this concern. We acknowledge that the mitigation potential of some policy-relevant but non-quantifiable targets is not fully captured in our modelling framework. However, we do not expect our focus on 47 quantifiable targets to materially overstate mitigation contributions from easily measurable energy-related sectors (e.g., power and transport) in a way that would change the main conclusions. IEA 2023 data indicate that around 96% of China’s CO₂ emissions originate from energy-related sectors, including electricity and heat production, transport, industry, and buildings (residential, commercial, and public services), which are largely covered by our policy assessment and the TIAM-UCL modelling scope. Therefore, the implications of excluding targets related to water, waste, recycling, air pollution control, and climate finance are unlikely to materially affect our findings on China’s overall energy transition and future CO₂ emissions, although a more comprehensive assessment of these non-energy measures remains an important direction for future work.

To improve transparency, we added clarifications on the potential selection bias and its likely direction in both Discussion and SI.

In Discussion lines 268-275:

“However, due to the quantification limits of TIAM-UCL, targets related to water, waste, recycling, air pollution, and climate finance are excluded from our modelling (see SI Table 2). Nonetheless, the 47 modelled targets cover all major energy-related sectors, including electricity and heat production, transport, industry, and buildings. Together, these sectors account for around 96% of China’s CO₂ emissions, based on IEA 2023 data²⁰. Therefore, any underestimation arising from the excluded targets is unlikely to materially affect our conclusions on China’s energy transition and future carbon emissions, although assessing these effects more comprehensively is an important direction for future research.”

11. The first Results subsection (“Chinese Decarbonization and Energy Transformation”) presents outcomes of China’s energy system transformation under different domestic policy scenarios (GW–GL) without explicitly stating the assumed global context. Please specify which global context is used for Fig. 1 and briefly note whether the qualitative conclusions are robust across both contexts. Nevertheless, a brief clarification in the text—stating that these results are “global context–neutral” or not materially affected by ROW assumptions—would help avoid potential confusion for readers who might otherwise expect Figure 1 to correspond to a specific global scenario.

Response:

Thank you for pointing this out. We agree that clarifying the global context for Figure 1 is important. To maintain a coherent flow of the results subsection, we added this clarification in the Figure 1 caption (Lines 180-183):

“It should be noted that China related modelling results presented in Figure 1 are based on the global NDC context. However, due to the underlying TIMES model structure, China’s domestic energy transition is primarily driven by end-use demand and domestic policy constraints and is not materially affected by assumptions about the rest of the world.”

Reviewer #2 (Remarks to the Author):

I reviewed an earlier version of this manuscript and found it to be a strong contribution, pending revisions. I have now reviewed the revised version and the authors’ responses to reviewers. I am satisfied that the authors have adequately addressed the concerns raised in the initial round of review. Their revisions are appropriate and improve the clarity and robustness of the manuscript. I recommend the manuscript for publication.

Response:

Thank you very much for your positive assessment of the revised manuscript. We are glad that the revisions have adequately addressed the reviewer’s concerns and improved the overall clarity and robustness of the work. We appreciate the reviewer’s recommendation for publication.